# Multiplexed expansion revealing for imaging multiprotein nanostructures in healthy and diseased brain

Jinyoung Kang[1,2,14], Margaret E. Schroeder [1,3,14], Youngmi Lee[1], Chaitanya Kapoor [4], Eunah Yu[1], Tyler B. Tarr[5], Kat Titterton[1], Menglong Zeng[1], Demian Park[1], Emily Niederst[6], Donglai Wei[7], Guoping Feng [1,2,3,8] & Edward S. Boyden [1,2,3,9,10,11,12,13]

Proteins work together in nanostructures in many physiological contexts and disease states. We recently developed expansion revealing (ExR), which expands proteins away from each other, in order to support better labeling with antibody tags and nanoscale imaging on conventional microscopes. Here, we report multiplexed expansion revealing (multiExR), which enables high-fidelity antibody visualization of >20 proteins in the same specimen, over serial rounds of staining and imaging. Across all datasets examined, multiExR exhibits a median round-to-round registration error of 39 nm, with a median registration error of 25 nm when the most stringent form of the protocol is used. We precisely map 23 proteins in the brain of 5xFAD Alzheimer's model mice, and find reductions in synaptic protein cluster volume, and co-localization of specific AMPA receptor subunits with amyloid-beta nanoclusters. We visualize 20 synaptic proteins in specimens of mouse primary somatosensory cortex. multiExR may be of broad use in analyzing how different kinds of protein are organized amidst normal and pathological processes in biology.

A single cell contains perhaps thousands of kinds of protein, which interact over nanoscale distances with each other to mediate biological processes. Disturbing that arrangement can corrupt signaling, and lead to pathological states[1]. Ideally one would be able to map the location, and identity, of many proteins within the same preserved cell or tissue specimen, with nanoscale precision. Such a map could generate novel hypotheses, and even insights, into how proteins might interact with each other, in a healthy or diseased state. However, studying proteins in their system contexts is complex, due to their

nanoscale size, and the crowded nature of their biological environment. Expansion microscopy is a form of light microscopy that benefits from physical expansion of specimens, via chemical introduction of a densely permeating swellable hydrogel throughout a biological sample. Biomolecules or labels of interest are covalently anchored to the hydrogel. The specimen is then chemically softened, and then water is added, causing the hydrogel-specimen composite to swell in an even fashion (typically 4x, although more recent protocols support 10x and 20x, and iterating the 4x procedure can also yield 20x). The

[1]McGovern Institute for Brain Research, MIT, Cambridge, MA, USA. [2]Yang Tan Collective, MIT, Cambridge, MA, USA. [3]Department of Brain and Cognitive Sciences, MIT, Cambridge, MA, USA. [4]Department of Electrical and Electronics Engineering, BITS Pilani, Rajasthan, India. [5]Department of Neuroscience, University of Pittsburgh, Pittsburgh, PA, USA. [6]The Picower Institute for Learning and Memory, MIT, Cambridge, MA, USA. [7]Department of Computer Science, Boston College, Chestnut Hill, MA, USA. [8]Broad Institute of MIT and Harvard, Cambridge, MA, USA. [9]Center for Neurobiological Engineering and K. Lisa Yang Center for Bionics, MIT, Cambridge, MA, USA. [10]Department of Biological Engineering, MIT, Cambridge, MA, USA. [11]Koch Institute, MIT, Cambridge, MA, USA. [12]Howard Hughes Medical Institute, Cambridge, MA, USA. [13]Media Arts and Sciences, MIT, Cambridge, MA, USA. [14]These authors contributed equally: Jinyoung Kang, Margaret E. Schroeder. ✉e-mail: edboyden@mit.edu

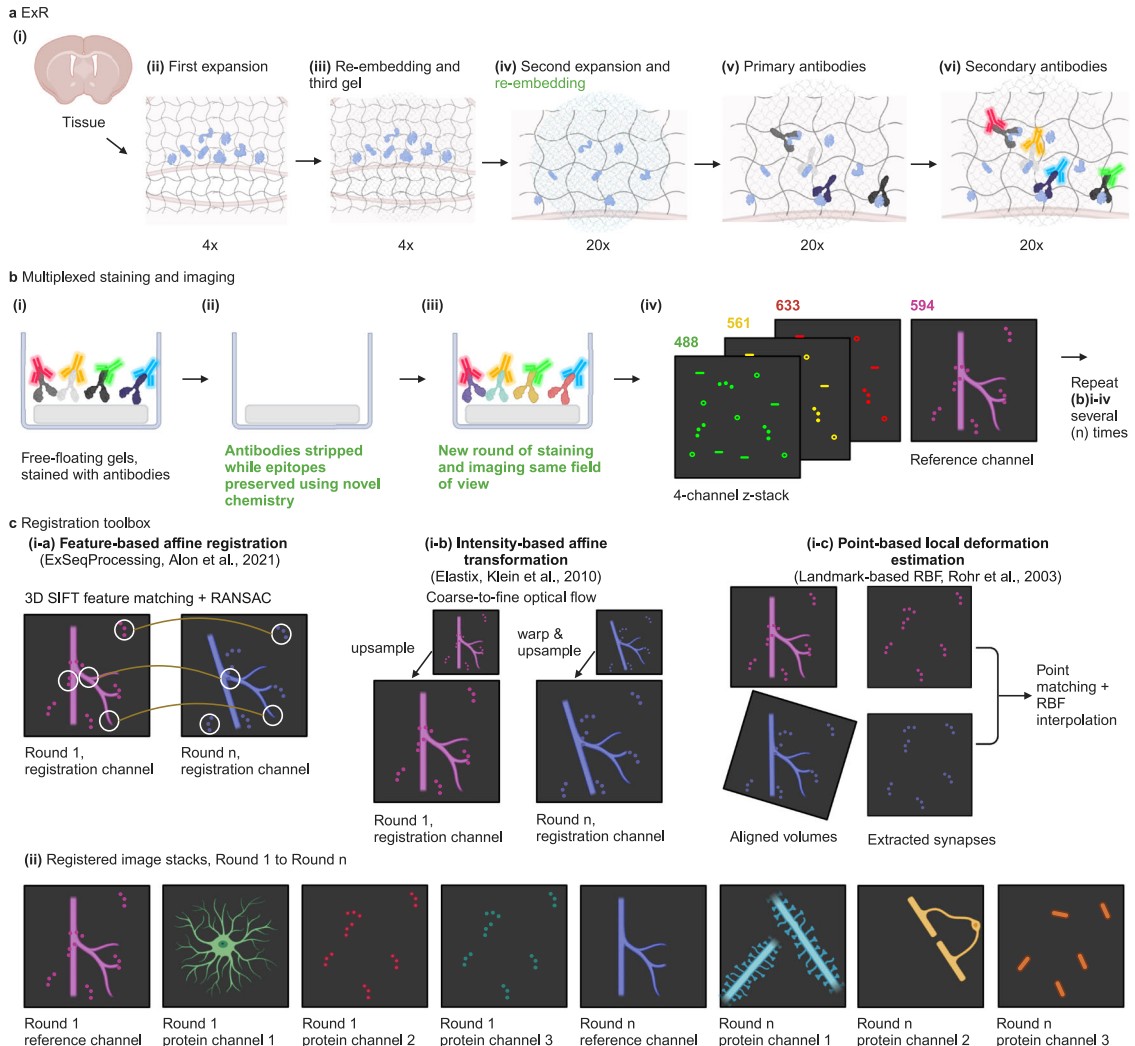

**Fig. 1 | Schematic of multiExR procedure. a** Expansion revealing (ExR), a technology for decrowding of proteins through isotropic protein separation. **a**i Coronal section of mouse brain before staining or expansion. **a**ii Anchoring and first gelation step. The specimen is embedded in a swellable hydrogel (gray wavy lines), mechanically softened via detergent and heat treatment, and expanded in water. **a**iii Re-embedding and second swellable gel formation. The fully expanded first gel is re-embedded in a charge-neutral gel (not shown), followed by the formation of a second swellable hydrogel (light gray wavy lines). **a**iv Final up to 20x expansion with the addition of water, followed by a recommended re-embedding step to preserve gel strength for multi-round imaging (blue wavy lines). **a**v, Post-expansion primary antibody staining (Y-shaped proteins). **a**vi Post-expansion staining with fluorescent secondary antibodies to visualize decrowded biomolecules. **b** Multiplexed ExR procedure. **b**i Free-floating gels are stained with conventional primary and secondary antibodies, and the images are collected. **b**ii After imaging, primary and secondary antibodies are stripped using detergent and heat-based denaturation while endogenous proteins are preserved by physical anchoring in

hydrogel networks. **b**iii Gels are re-incubated with a new round of primary and secondary antibodies, and the same field of view is imaged again. **b**iv A 3 or 4-channel z-stack is obtained on a confocal microscope. One or more of the four channels serves as the reference channel. After imaging, the antibody stripping and staining processes are repeated for up to 10 rounds. **c** Registration of multi-round images using the reference channel. The multi-round images are registered using one or a combination of the methods (i-**a** and i-**c**, or i-**b** and i-**c**) in this toolbox (see Supplementary Fig. 1 and "Methods" section for more details). i-**a** a feature-based affine registration algorithm[8,9]. i-**b** an intensity-based affine registration algorithm[10] iteratively refining the estimation from the coarse scale of the image pairs to the fine scale. i-**c**, a point-based registration algorithm[11], designed specifically to further align fine structures. **c**ii Registered multiExR images are obtained after applying calculated warps to all channels from later rounds, creating multi-channel image volumes. Schematic created with BioRender.com. Bolded, green text highlights technical innovations of the multiExR procedure.

net effect is that the light microscope has an effectively increased resolution, even beating the diffraction limit[2].

Early forms of expansion microscopy focused on labeling biomolecules before expansion[3]. Recently, forms of expansion microscopy such as expansion revealing (ExR[4], Fig. 1a), which pulls proteins apart from each other throughout a specimen via a process of specimen hydrogel embedding, protein anchoring to the hydrogel, epitope-preserving specimen softening, and isotropic sample swelling, have begun to enable densely packed proteins to be separated from one another. Separated proteins can be more easily stained by labels, given the better access supported. Thus, proteins previously invisible in light

microscopy can become visible. ExR, which expands samples by ~20x in linear dimension, supports ~20 nm resolution imaging on ordinary microscopes. To date, ExR has been used to visualize a few proteins at once within a specimen, limited by the spectral properties of the fluorophores used. Many different hardware platforms have been proposed for the purposes of improving the number of biomolecules visualizable within a specimen[5–7], but most of these are not commonly available in ordinary biology laboratories. Techniques like ExM have become popular in part because they do not require novel hardware to be purchased by a biology group, which has led to rapid adoption in everyday biology experiments[3]. We thus asked whether it would be

possible to devise a multiplexed form of ExR that could enable the imaging of potentially arbitrary numbers of proteins, with nanoscale precision, in the same expanded specimen, without requiring hardware not commonly available in biology labs.

We here describe a multiplexed form of ExR (multiExR), which extends ExR through serial rounds of post-expansion staining, imaging, and washing. We optimized and validated each chemical step of this process, enabling ~20 proteins in the same tissue sample to be visualized, using conventional antibodies, with low signal deterioration or bleed-through between rounds. Due to the mechanical properties of expanded gels, precise registration of these images across rounds posed a challenge. We optimized experimental conditions, and building from prior work[8], implemented registration algorithms to register serial imaging rounds with high precision. In the most stringent form of the protocol (Supplementary Fig. 1Bii, path 2 in Supplementary Fig. 1A), which we applied to our primary validation dataset, we achieved a median (taken across all staining round pairs for all fields of view, i.e. each combination of round pair and field of view was considered as one sample) round-to-round registration error of 25 nm (minimum 14 to maximum 98 nm). The staining and registration steps can be tuned to trade off between precision and throughput, as prioritized by the user. We do not recommend the aforementioned stringent form for beginners, because it is laborious: if such high precision is not required, the process can be conducted more quickly, albeit at the cost that the round-to-round alignment error could rise to 100 nm or more.

We demonstrated that multiExR, used with 23 different antibodies against different proteins, could be used to characterize known, and previously undescribed, pathology, revealing nanoscale colocalizations between multiple synaptic proteins and amyloid-beta (Aβ) nanoclusters in the 5xFAD Alzheimer's model mouse brain. We also visualized putative synapses, imaging 20 proteins in the same specimen, in the mouse somatosensory cortex. Thus, multiExR offers great utility in the mapping of protein organization in healthy and disease states, potentially yielding novel hypotheses of molecular mechanism and/or drug target, and perhaps even someday diagnostics, in biology and medicine.

## Results

### Optimizing ExR for multi-round staining, imaging, and registration

The approach for multiExR is schematized in Fig. 1. Tissue and gel preparation are identical to those of ExR, until after the final expansion step, when an additional re-embedding step is performed to increase gel density and strength (Fig. 1a). One channel serves as a reference channel to enable registration across imaging rounds. For most of the datasets presented here, we used three different molecular targets, all labeled with the same fluorophore, to provide a reference channel that exhibits features ranging from the nanoscale to the macroscale, so that accurate registration is possible both at the scale of the entire specimen, as well as at the nanoscale. Criteria for a protein (or set of proteins) to be chosen as a reference include high signal-to-noise ratio and adequate feature density. It may also be useful to choose a reference protein with some structural, morphological, or cell type information that guides the viewer to appropriate subvolumes for detailed imaging. In the current study, we primarily use Lycopersicon Esculentum Lectin combined with glial fibrillary acidic protein (GFAP) and neurofilament (SMI) and sometimes the synaptic scaffold Homer1, with all molecules being labeled with the same color fluorophore, as a reference channel. The multi-scale nature of our reference channel is important, since a reference channel with only macroscale features lacks the information to support registration with nanoprecision, and a reference channel with only nanoscale features is difficult to align at the macroscale.

During imaging, we searched for fields of view with high feature density in the reference channel, to ensure adequate feature density in the reference channel used to support alignment, for each field of view. More specifically, we found that a reference channel feature density of ~1.3% (Supplementary Table 1), with both large (e.g., largest feature ~50% of total reference feature volume like Lectin, Supplementary Table 1) and small (e.g., smallest features $2.832^{-3}$% of total reference feature volume like synaptic proteins, Supplementary Table 1) features present (Supplementary Fig. 2a), was sufficient for accurate registration across rounds with median round-to-round precision of 25 nm across all staining round pairs for all fields of view, in the most stringent form of the protocol (Supplementary Fig. 1Bii). Here, and throughout the paper, unless otherwise indicated, this median was calculated as follows: first, registration errors were averaged across ~1,000 randomly sampled subvolumes within each field of view. Then, those averages were analyzed across round pairs (e.g., rounds 1–2, 1–3, 1–4, etc.) for all fields of view, and the median taken across all round pairs and fields of view. We calculated feature density as the fraction of total image volume occupied by feature volume in the reference channel, which is the normalized sum of all channels in that round in Fig. 2, but in other datasets is the three-target channel described above. The staining, imaging, and registration steps can be relaxed in stringency if such precision is not needed, but greater throughput is valued, resulting in a version of the protocol that offers usually <100 nm, but occasionally higher, round-to-round precision (see Supplementary Fig. 1 for flowchart to help users choose the optimal protocol for a given scientific question). We recommend this relaxed-stringency protocol for beginners, due to its simpler nature.

Following the first round of imaging, antibodies are stripped using an ExR-optimized protocol. 100 mM beta-mercaptoethanol-containing denaturation buffer (200 mM sodium dodecyl sulfate (SDS), 200 mM NaCl, and 50 mM Tris pH 9), which can break antibody disulfide bonds and support protein denaturation with minimal heat treatment, was used to remove antibody stains post-imaging (Fig. 1b, see "Methods" section for details). These stripping conditions were optimal for preserving anchored epitopes while minimizing signal carryover between staining rounds, as shown below. The next round of staining, imaging, and stripping is then performed, iterating as needed. The reference channel, which in most of the datasets presented here was composed of three or more different molecular targets (Supplementary Fig. 1), is used to locate the same field of view in each imaging round, and to align the images across rounds, as described in the next paragraph. Importantly, to minimize gel drift, for successful downstream registration, we found it necessary to stabilize gels before taking images, by removing the buffer around gels as much as possible, and placing a sealing film over the well plate containing the gel to avoid gel drying during imaging.

Computational registration with nanoscale precision of multiExR images taken across multiple rounds of staining and imaging initially posed great difficulty, despite innovations such as our multiscale reference stain, and the aforementioned improvements in sample stabilization. multiExR gels are free-floating, because immobilization of gels used in standard ExM imaging reduces antibody stripping efficiency (and gels would often detach from glass surfaces during antibody stripping). However, free-floating gels exhibit more degrees of freedom, and thus variability, between rounds of imaging than immobilized gels. Furthermore, due to the highly expanded nature of ExR gels, and the dilution of tissue structure due to such expansion, features can be sparser than ideal for registration. Finally, slight variation in signal-to-noise ratio (SNR), perhaps due to the stochasticity of antibody binding visible at the nanometer scale, means that even identical staining and imaging conditions across rounds can lead to slightly, but significantly, different images in the reference channel – perhaps a fundamental issue for any nanoimaging protocol involving antibody staining. For these reasons, multiExR registration cannot be accomplished with conventional intensity-based methods optimized

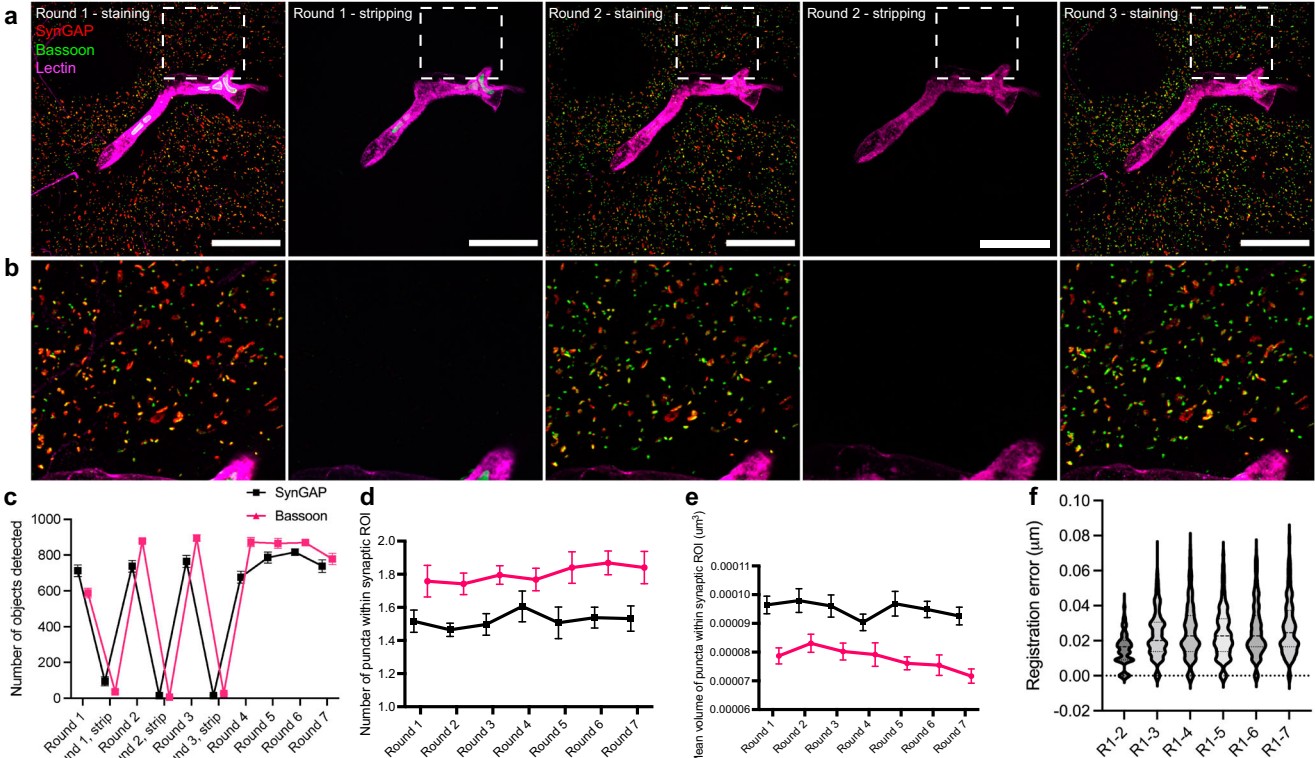

**Fig. 2 | Validation of multiExR technology by staining, stripping, and re-staining the same set of primary and secondary antibodies across multiple rounds in the mouse somatosensory cortex. a** Example field of view (max intensity projection) of registered validation dataset images in round 1, stripping after round 1, round 2, stripping after round 2, and round 3. Pixel intensities are adjusted to the same minimum and maximum values for staining and stripping rounds. **b** Zoom in of boxed region of (**a**). Scale bar, 5 μm in biological units (i.e., real size divided by expansion factor). **c** Mean number of objects detected in a field of view (see "Methods" section) after 7 staining rounds and the first 3 stripping rounds ($n = 7$ fields of view from one mouse for staining rounds, where the first 3 stripping rounds were imaged but stripping was performed between all rounds). **d** Mean number of puncta detected in manually-identified synaptic regions of interest (ROIs) after 7 staining rounds (the same $n = 7$ fields of view from one mouse, mean is taken over 51-53 ROIs per field of view). **e** Mean volume of puncta detected in manually-identified synaptic ROIs after 7 staining rounds (the same $n = 7$ fields of view from one mouse, mean is taken over 51-53 ROIs per field of view). Error bars in **c–e** represent standard error of the mean across the fields of view. **f** Estimated population distribution (violin plot of density, with a dashed line at the median and dotted lines at the quartiles) of the registration error in a representative field of view (different from panels (**a, b**), as it was more representative of registration error). The 95% confidence interval for each round pair is [0.01467, 0.01578] for rounds 1–2, [0.02271, 0.02430] for rounds 1–3, [0.02443, 0.02635] for rounds 1–4, [0.02337, 0.02516] for rounds 1–5, [0.02491, 0.02881] for rounds 1–6, and [0.02657, 0.02855] for rounds 1–7 (see "Methods" section, $n = 1000$ randomly sampled subvolumes from one field of view from one mouse). Source data are provided as a Source Data file.

for fixed, unexpanded tissue images, such as those in popular Fiji plugins.

To address this challenge, we created a toolbox for both global and local alignment of multiExR imaging rounds, consisting of: a 3D scale-intensity feature transform (SIFT)-based global registration algorithm adapted from previous algorithms for multiround alignment of multiplexed RNA ExM images (ExSeqProcessing registration[8,9]), an alternative intensity-based global registration algorithm[10], and an optional point-based alignment step for refinement of local structures such as synapses[11] (Fig. 1c, see "Methods" section for details). Users can choose between the two global algorithms, and whether to add a local registration step, based on the signal-to-noise ratio of their images (see Supplementary Fig. 1 for a flowchart of how to choose the experimental and computational workflow depending on desired goal). If it is necessary to register small, punctate objects such as synapses to one another, with high accuracy, across subsequent rounds, the point-based alignment algorithm can improve local registration accuracy.

We recommend starting with the feature-based ExSeqProcessing registration algorithm, as we found this could accurately and reliably align multiExR imaging rounds for most of our datasets within 100 nm (median 43 nm, minimum 6.2 nm, maximum 151 nm) and has a fast runtime of <30 minutes per field of view when implemented with graphic processing unit (GPU) acceleration. Users can quickly evaluate

registration quality by examining composite overlays of the reference channel between different imaging rounds in Fiji: micron-scale registration errors will be evident by a lack of colocalized signal. Nanoscale registration errors can be quantified in a few hours using the pipeline described in the "Methods" section.

We quantitatively validated multiExR signal, background, and registration error, by staining for the same target synaptic proteins (SynGAP and Bassoon) repeatedly over seven rounds of staining, imaging, and stripping, using the same microscope settings. In general, the absolute intensity of an immunostained protein can be highly variable and depends on many experimental factors, some of which are controllable and some of which are not. Therefore, we do not claim that absolute intensity is constant over multiple rounds of stripping and staining with multiExR. Instead, we focus on measures of volume, signal-to-noise, intensity ratio, and number of detectable objects – more robust metrics – in the following analyses.

To confirm stripping efficacy, we stained only with Lectin after stripping antibodies, to find the same field of view, then obtained images using the same laser power and exposure time, finding that there was minimal residual signal, which in turn would lead to negligible bleed-through between rounds (Fig. 2a, b and Supplementary Fig. 2). Later rounds maintained high SNR for target synaptic proteins, evidenced by cross-round stability in the number of detected objects

in the whole field of view (putative synapses, Fig. 2c and Supplementary Fig. 2a), the SNR of specific synaptic proteins within manually-identified putative synapses (Supplementary Fig. 2b), the number of protein puncta within these putative synapses (Fig. 2d) and the volume of these puncta (Fig. 2e). We note that the last three of these measures compare the SNR and number of puncta within identified putative synapses, relevant to the building up of information about a given synapse over many rounds of protein identification and localization. We note persistent Bassoon staining on the blood vessel after stripping round 1, but not after stripping round 2. We speculate that insufficient stripping may be more likely to occur for "stickier" structures like blood vessels, where there may be more non-specific binding of Fc fragments, as we did not observe insufficient stripping outside of blood vessels. We observed non-specific staining in blood vessels, for some proteins, in all datasets (Figs. 3a, b and 5a and Supplementary Figs. 3a and 6).

We observed that the Bassoon signal intensity, relative to SynGAP, increased markedly after the first round of stripping, and remained stable in subsequent rounds (Fig. 2a, b). To quantify this increase, we calculated the mean signal intensity (in background subtracted images) of pixels in synaptic puncta within manually-identified synaptic regions of interest (ROIs) and found this to be increased in the second and third staining rounds (Supplementary Fig. 2ci), consistent with an antigen retrieval-like effect following the harsh denaturation conditions used in antibody stripping, that affects the Bassoon but not SynGAP target epitope. In contrast, the absolute intensity of SynGAP staining, and of Bassoon staining on rounds beyond the third round, decreased somewhat steadily with successive rounds of stripping and re-staining (Supplementary Fig. 2cii), which suggests a general process of epitope staining efficacy decline occurs during harsh stripping conditions. However, the signal-to-noise ratio, number of puncta and puncta volumes (Fig. 2d, e and Supplementary Fig. 2b) were stable across rounds, demonstrating that while absolute intensity may vary between rounds of staining, detection of biologically meaningful objects was maintained. In particular, it is striking that although absolute signal drops by a factor of 2 or 3, signal-to-noise (computed as signal of the object divided by signal of the background) stays constant – meaning that changes in epitope staining efficacy may apply equally to background staining as to object staining, consistent with other work[12]. However, not all epitopes increase staining after stripping treatment; some decrease in brightness (Supplementary Fig. 2c). Thus, the cost and benefit of stripping-based antigen retrieval will need to be evaluated on a target-by-target basis, in pilot experiments, to gauge whether to pursue it deliberately or not, before staining.

To assess whether the stripping process affected the reference channel, we measured the mean signal intensity of the maximum intensity projection of the registered Lectin channel for each field of view and each staining and stripping round, in which the reference channel was re-stained to locate the same field of view as in the previous round. While the mean signal intensity in the single-channel Lectin reference channel did decline somewhat after the first round (Supplementary Fig. 2d and Supplementary Table 3i), the magnitude of the mean signal intensity was relatively stable in later rounds. Taken with the SynGAP and Bassoon findings, this result highlights that antigen retrieval effects may not only boost intensity, but may sometimes suppress intensity. While there was significant variation in mean signal intensity between the rounds, there was no systematic pattern to the variation (Supplementary Table 3ii–iii). Given that decreases in intensity are possible with multiExR, it is possible that multiExR will not be ideal in situations where signals are very weak, or single molecule counting is required; given that one key advantage of ExR is the decrowding of densely packed, concentrated protein clusters, for better labeling, this may be a moot point. Nevertheless, even in the stripping rounds, where only the Lectin channel was re-stained (Supplementary Fig. 1B(iii)), same as reference channel strategy 1(iii), using

registration algorithm 2(i)), there remained sufficient signal to register the images within 80 nm on average, and often within 50 nm on average (Supplementary Table 2). Thus, regardless of the magnitude of signal intensity in the reference channel, its functional integrity is maintained after stripping, to an extent sufficient to make meaningful conclusions and to align images.

For this first validation dataset, we utilized 4 single-protein channels combined (normalized and summed) to a single reference channel (path 7, reference channel option 1(ii) in Supplementary Fig. 1, omitting registration algorithm 2(iii)). The median registration error across all staining round pairs for all fields of view, calculated as previously described[8], using the SynGAP channel, was 25 nm (i.e., 2–3 pixels for the microscope settings used), with consistent performance for most round pairs: 71.4% exhibited registration error <30 nm (Fig. 2f, see Supplementary Table 2 for full statistics).

In a second validation experiment (Supplementary Fig. 3), we used 4 proteins (Lectin, neurofilament SMI, GFAP, and Homer) in a single reference channel for registration (path 9, reference channel option 1(i) in Supplementary Fig. 1), similar to what we used in many later experiments (Figs. 3–5), to increase the number of target proteins that can be imaged per round. Four round pairs that had poor or failed registration using the global feature-based registration algorithm (2(i) in Supplementary Fig. 1c) were registered using the global intensity-based algorithm (2(ii) in Supplementary Fig. 1c). The median registration error, calculated as described above using the reference channel, was 66 nm, with consistent performance for most staining round pairs (59.3% of staining round pairs for all fields of view exhibited mean (across subvolumes) registration error <70 nm) in most fields of view, and all but one round pair across all fields of view had mean (across subvolumes) registration error <100 nm in staining rounds (Supplementary Fig. 3b, see Supplementary Table 4 for full statistics). We speculate that the higher registration error in this secondary validation dataset was due to signal intensity differences between the proteins co-stained in the single reference channel (Supplementary Fig. 3a), which reduced the quality of feature detection across scales during SIFT-based registration. The registration accuracy obtained from rounds using the global intensity-based algorithm, which was only used when the SIFT-based algorithm failed, was similar to that of the other rounds (Supplementary Table 4).

Seven rounds of multiExR could reveal 21 different biomolecules, if three target proteins were imaged per round, with the fourth color reserved for the reference channel. We demonstrate at least 20 proteins in the same field of view, in the examples in the rest of this paper. To our knowledge, this is the highest number of different biomolecules visualized in the same tissue specimen with such nanoscale spatial precision. In principle, 20-protein multiExR could reveal 20 ×19 / 2 = 190 different protein-protein relationships.

## Nanoscale multiplexed characterization of amyloid beta and synapse pathology in Alzheimer's model mouse brain

To demonstrate the utility of multiExR for profiling multiplexed protein configurations, we explored the nanoscale organization of 23 proteins in 12-month-old Alzheimer's model 5xFAD[13] and age-matched wild-type (WT) mouse brains (12-13 months of age; Fig. 3a, b and Supplementary Table 5). We chose to characterize three amyloid-beta species with antibodies: 12F4[14], which targets the C-terminus of Aβ−42; D54D2, which targets several isoforms of human Aβ (Aβ−37, Aβ−38, Aβ−39, Aβ−40, and Aβ−42); and 6E10, which targets amino acid residues 1-16 of human Aβ, to see how their staining patterns differed, with nanoscale precision. For registration across rounds, we used registration channel option 1(i) and registration algorithm 2(i) from Supplementary Fig. 1C. Images were obtained from 8 fields of view from 2 WT mice and 9 fields of view from 2 5xFAD mice. We achieved a median registration error of 34 nm across all fields of view and all round pairs (Supplementary Table 6).

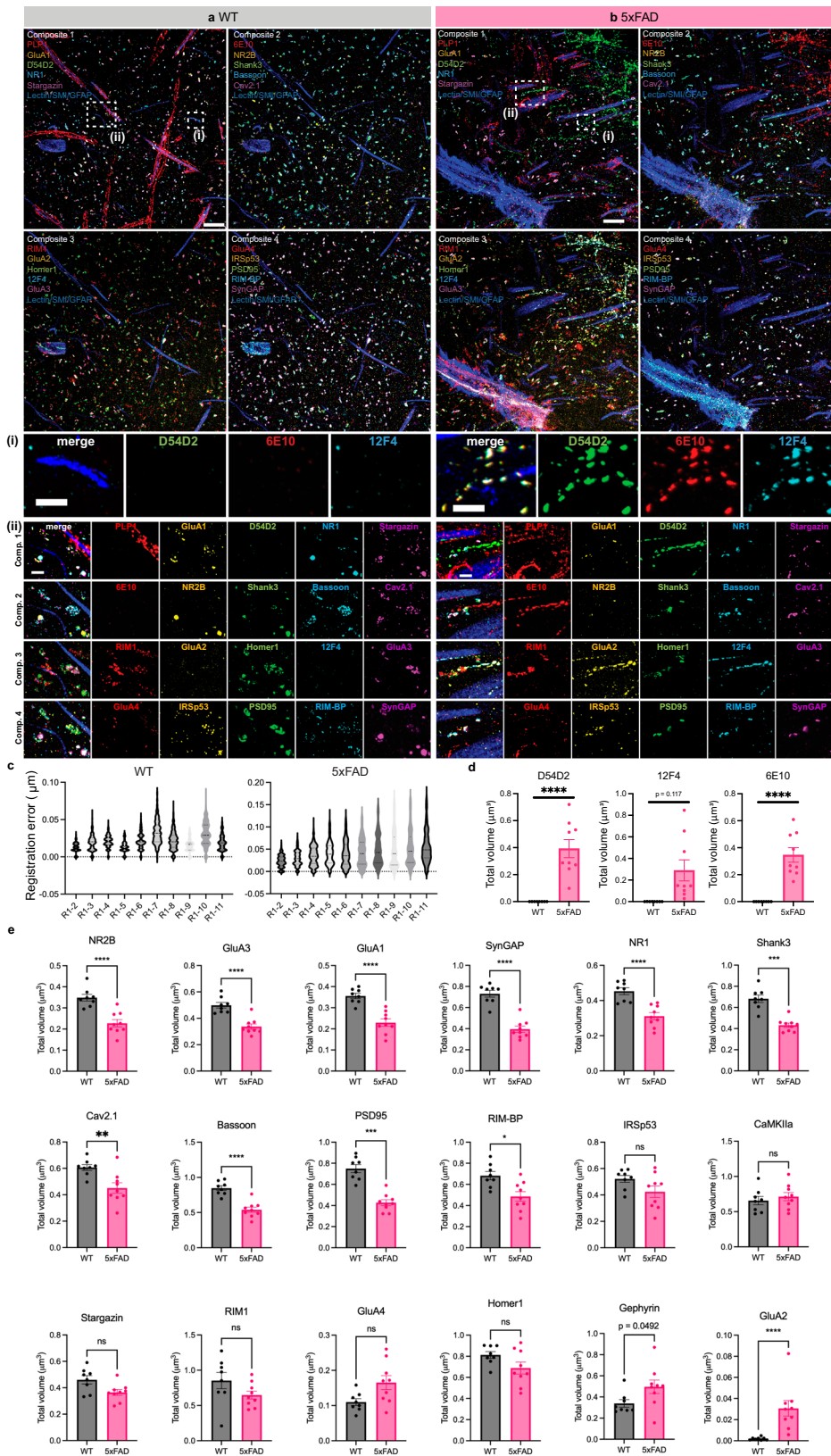

As expected, we observed increased amyloid beta burden as measured by the total volume of intensity-thresholded regions in the cortex of 5xFAD brains as compared to WT for D54D2 and 6E10 (Fig. 3d). The difference in volume for D54D2 and 6E10 reached statistical significance (linear mixed effect model accounting for multiple observations per animal, p-value on group effect = $1.19 \times 10^{-5}$ for D54D2, 0.117 for 12F4, and $1.50 \times 10^{-8}$ for 6E10). We then leveraged multiplexing to examine synapse alterations, a hallmark of Alzheimer's disease pathology that precedes cognitive decline[15], at nanoscale resolution. We first wondered whether Aβ nanodomains might preferentially colocalize with specific synaptic proteins in 5xFAD brains (Fig. 3bii), and proceeded to analyze the data in several steps. First, we

**Fig. 3 | 23-plex nanoscale characterization of amyloid beta pathology and synapse loss in Alzheimer's model mouse somatosensory cortex. a, b** 6-channel and composite maximum intensity projections of Aβ and synaptic proteins in representative fields of view and zoom-ins (lower panels) from WT (**a**) and 5xFAD (**b**), obtained using multiExR. Scale bar, 2 μm (upper panels), (i) and (ii) 500 nm. **c** Violin plots of the population distribution of registration error for these fields of view. **d** Total volume in intensity-thresholded regions (see Methods) for D54D2, 12F4, and 6E10 Aβ species in WT and 5xFAD registered fields of view (statistical significance determined using a linear mixed effects model without multiple

comparisons correction, $n = 17$ fields of view from two WT and two 5xFAD animals, error bars are mean ± standard error of the mean. **e** Total volume of objects detected after intensity thresholding and size filtration in WT and 5xFAD registered fields of view (statistical significance determined using a linear mixed effects model without multiple comparisons correction, the same $n = 17$ fields of view from two WT and two 5xFAD animals, error bars are mean ± standard error of the mean) for various synaptic proteins (see Supplementary Table 7 for full statistics). WT wild type, 5xFAD 5x familial Alzheimer's disease model mice. Source data are provided as a Source Data file.

examined the 18 synaptic proteins we stained, at locations that did not exhibit amyloid, which might represent proteins at putative synapses that do not contain amyloid (which might be in a different physiological or health state than synapses that did contain amyloid). We masked out Aβ, and then quantified synaptic proteins that were non-overlapping with such amyloid hotspots after additional median and size filtering (see "Methods" section). 10 out of the 18 proteins we thus analyzed were significantly decreased in 5xFAD brains (Fig. 3e, with statistical significance determined using a linear mixed effects model, $n = 8\text{-}9$ fields of view per group from 2 animals per group, see Supplementary Table 7 for full statistics). One out of the 18 proteins, GluA2, was significantly increased in 5xFAD mice compared to wild-type, where GluA2 expression visually appeared quite sparse at this age timepoint, at least relative to the higher levels seen in 5xFAD mice (Fig. 3b(ii)).

We next examined synaptic proteins that co-localized with Aβ nanoclusters, structures that cannot be observed with diffraction-limited confocal microscopy of non-expanded samples, or even with pre-expansion staining of samples that are then expanded[4]. Thus, visualization of such nanoclusters requires the epitope decrowding effect afforded by ExR and multiExR. We manually identified Aβ nanocluster ROIs in which all three Aβ stains were positive, and counted the total volume of puncta for each synaptic and Aβ channel contained in each ROI. We observed a relatively high volume occupied by GluA2 and CaMKIIa in these nanocluster ROIs (Fig. 4a, b). In contrast, the volume occupied by other synaptic proteins and PLP1 (which we added to this analysis, after visually observing it to colocalize with amyloid nanoclusters) contained within these Aβ nanocluster ROIs was smaller (see Supplementary Table 8 for descriptive statistics). Given the prominence of GluA2, we wondered if other α-amino-3-hydroxy-5-methyl-4-isoxazolepropionic acid receptor (AMPAR) subunits, i.e. GluA1, GluA3, and GluA4, were also present; GluA1 was found at trace amounts in these puncta, in terms of volume occupied, on average, GluA3 was present in a fraction of puncta, but GluA4 was present more consistently (Fig. 4b).

Given that AMPA subunits assemble in heteromeric fashion to form functional receptors, we wondered if there was any relationship between the amount of GluA2 within a Aβ nanocluster, and the amount of GluA4 found therein. Because all Aβ species imaged exhibited qualitatively similar staining patterns (Fig. 3), we arbitrarily chose to use D54D2, for this analysis, to quantify synaptic protein colocalization with amyloid. By calculating the volume of D54D2 that contained each AMPAR subunit as the intersection of GluA1/2/3/4 volume and D54D2 volume divided by total D54D2 volume, we found a significantly larger fraction of D54D2 containing GluA2 than GluA4 (Fig. 4c, $n = 44$ nanocluster ROIs from 8 fields of view from 2 5xFAD mice, Supplementary Table 9(i–ii) for statistics). We found essentially no GluA1 and very little GluA3 within D54D2 puncta (Fig. 4c). Additionally, there were many more ROIs for which there was no GluA4 contained within a D54D2 punctum, than for GluA2 (Fig. 4c, 4.55% zero for GluA2 vs. 52.3% for GluA4; Chi-squared test, Chi-square = 24.64, $p < 0.0001$, $n = 44$ nanocluster ROIs from 8 fields of view from 2 5xFAD mice). Leveraging the multiplexed nature of the data, we performed pairwise linear regressions on the volume of GluA4 and GluA2 vs. D54D2 present in Aβ nanocluster ROIs, and found that both were highly correlated, but the

best-fit line for GluA2 vs. D54D2 volume was shifted up from that of GluA4 vs. D54D2 (Fig. 4d and Supplementary Table 9(ii) for full statistics). For Aβ nanocluster ROIs in which GluA4 was present, the volume of GluA2 present was correlated with that of GluA4 (Fig. 4e and Supplementary Table 9(iii) for full statistics). Visual inspection of GluA2 and GluA4 in Aβ ROIs chosen from different parts of the distributions of Fig. 4c illustrate these observations (Fig. 4f).

These results suggest non-random colocalization of GluA2, and to a lesser extent, GluA4, with Aβ nanodomains in the 12-month 5xFAD mouse brain. This colocalization illustrates the power of multiExR: given that most other synaptic proteins, and especially the other AMPAR subunits examined, GluA1 and GluA3, did not exhibit such striking colocalization with Aβ nanodomains, analyzing such pairwise and multi-way protein nanoscale coordination benefits greatly from being able to visualize many proteins in the same sample, with nanoscale precision. To our knowledge, this is the first observation of AMPAR aggregation in Aβ nanodomains in the 5xFAD mouse model, and may reflect pathological aggregation of these synaptic proteins in the context of amyloid pathology. AMPARs aggregate for synapse formation, and specific subunits affect AMPAR permeability and function. Co-aggregation of subunits with amyloid may have synaptic implications, either from altered AMPAR function or gross dysfunction.

### Nanoscale multiprotein visualization of synapses

We visualized, in wild-type mice, 20 synaptic proteins important for neural architecture and transmission: the presynaptic proteins bassoon, RIM1, RIM-BP, Vglut1; the P/Q-type Calcium channel Cav2.1 alpha 1A subunit (Cav2.1); the postsynaptic scaffold proteins Shank3, SynGAP, PSD95, IRSp53, Elfn1; neurotransmitter receptor subunits GluA1, GluA2, GluA3, GluA4, NR1, and NR2B; the AMPA receptor auxiliary subunit Stargazin; the calcium/calmodulin-dependent protein kinase II alpha subunit (CaMKIIa); gephyrin, a GABA (gamma-aminobutyric acid)-ergic synapse scaffolding protein; and the tyrosine kinase receptor protein Erbb4 (Fig. 5 and Supplementary Table 10). As we previously showed[4], some of these proteins cannot be visualized well with pre-expansion staining, due to the crowded nature of synapses, but can be easily visualized with ExR because decrowding proteins facilitates their staining with conventional antibodies.

Associated registration errors, with median of 45 nm across all rounds and fields of view in 2 mice, are provided in Supplementary Table 11. This error is consistent with, and indeed a bit smaller than, that of our secondary validation dataset, which used the same reference channel and registration strategy (path 9, reference channel option **1(i)** in Supplementary Fig. 1). One round pair that failed registration using the global feature-based registration algorithm (**2(i)** in Supplementary Fig. 1c) was registered using the global intensity-based algorithm (**2(ii)** in Supplementary Fig. 1c). Images were obtained from 8 fields of view from 2 mice (Supplementary Tables 10 and 11).

Many proteins overlapped or appeared adjacent to each other, as would be expected at synapses; the composition and expression level would vary from synapse to synapse. For example, postsynaptic scaffold proteins (SynGAP, PSD95, Shank3, Homer1) colocalized with each other (see line-headed white arrows in Fig. 5) and formed sandwich-like structures between pre- (Bassoon, RIM1) and post-synaptic scaffold

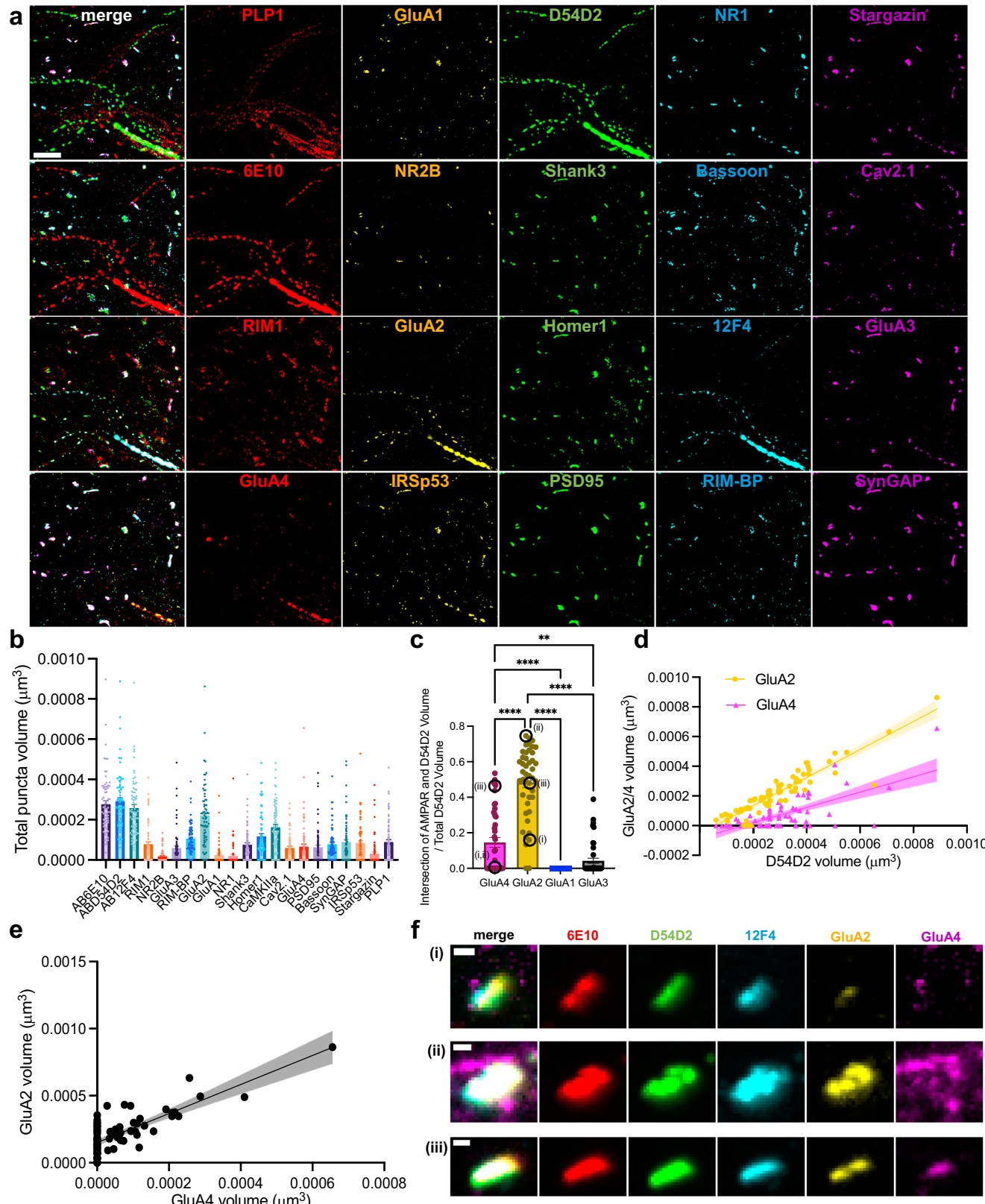

proteins (PSD95, SynGAP) (see triangle-headed white arrows in Fig. 5, zoomed into in **iii-iv**). AMPA receptor subunits (GluA1/3/4) and a transmembrane AMPA receptor regulatory protein (Tarp gamma-2, Stargazin) co-localized with each other (see blue arrows in Fig. 5). Gephyrin, a known inhibitory synaptic marker, was observed in some putative synapses with excitatory synaptic proteins nearby (AMPA

receptor, N-methyl-D-aspartate (NMDA) receptor, Shank3, Stargazin, SynGAP) (see red arrows in Fig. 5). VGlut1 was more scattered than the synaptic scaffold proteins we imaged in the same volume. We observed that VGlut1 signal was relatively more diffuse compared to these other proteins after maximum intensity projection of the 3-dimensional image volume. However, examining a single z-plane

**Fig. 4 | Analysis of nanoscale colocalization of synaptic proteins and amyloid-beta in Alzheimer's model mouse brain. a** Example 5-channel and composite maximum intensity projections of a 5xFAD field of view, cropped to show Aβ nanoclusters. (Scale bar, 1 μm). **b** Bar plots of total volume of select proteins within Aβ nanocluster ROIs ($n = 71$ ROIs from 9 fields of view from 2 5xFAD animals; Supplementary Table 8 for full statistics, error bars indicate mean ± standard error of the mean). **c** Bar plots of the fraction of volume of D54D2 occupied by AMPA receptor (error bars are mean ± standard error of the mean, statistical significance determined by Tukey's multiple comparisons test following one-way ANOVA, $p < 0.0001$ for all asterisked comparisons except $p = 0.0047$ for GluA3 vs. GluA4, $n = 44$ nanocluster ROIs from 8 fields of view from 2 5xFAD animals; Supplementary Table 9(i) for full statistics). **d** Scatterplot of GluA2 (yellow circles) and GluA4 (blue triangles) volume vs. D54D2 volume within Aβ nanocluster ROIs. Lines indicate the best-fit lines from simple linear regressions, and the shaded regions indicate the 95% confidence interval on the best-fit line ($n = 71$ ROIs from 9 fields of view from 2 5xFAD animals; Supplementary Table 9(ii) for full statistics). **e** Scatter plot of GluA2 volume vs. GluA4 volume within Aβ nanocluster ROIs. Black line indicates the best-fit line from a simple linear regression, and the shaded region indicates the 95% confidence interval on the best-fit line (the same $n = 71$ ROIs from 9 fields of view from 2 5xFAD animals; Supplementary Table 9(iii) for full statistics). **f** Maximum intensity projections for selected channels of the ROIs circled in black in the plot in **c**. Scale bar, 50 nm. ****$p < 0.0001$, ***$p < 0.001$ **$p < 0.01$, ns, not significant. WT, wild type. 5xFAD, 5x familial Alzheimer's disease model mice. The 5xFAD data are from the same animals and fields of view as Fig. 3. Source data are provided as a Source Data file.

revealed vGlut1 colocalization with Bassoon and SynGAP, as expected (Supplementary Fig. 5). In the future, detailed analysis of multiExR synaptic data could be useful for investigators seeking to characterize synapse types and states from a heterogenous synapse population.

We also demonstrate the use of multiExR to profile putative synapses in cultured neurons, as in vitro models are widely used in neuroscience. We imaged 10 synaptic proteins over 5 staining rounds (Supplementary Table 12): Synapsin1, NR1, NR2B, SynGAP, GluA1, PSD95, Bassoon, Gephyrin, RIM1, and CaMKIIa. We observed diffuse (non-punctate), large-volume Synapsin1 immunoreactivity, colocalization of postsynaptic proteins NR2B, SynGAP, NR1, and PSD95, and very sparse and dim gephyrin immunoreactivity (Supplementary Fig. 4a, b). For registration across rounds, we stained for SMI, GFAP, and Homer in a single reference channel (path 1, reference channel option **1(ii)** in Supplementary Fig. 1), used the global feature-based registration algorithm, and achieved a median registration error of 49 nm across all fields of view and round pairs (Supplementary Fig. 4c and Supplementary Table 13). This result demonstrates that multiExR can be used to quantitatively profile nanoscale structures in cultured neurons.

## Discussion

We here show that we can map many proteins in the same field of view, in the same intact specimen, with nanoscale precision, with multiExR. MultiExR builds from the high spatial resolution of ExR (~20 nm within a round (i.e., regular ExR, as previously published) with a median cross-round registration error of 39 nm across all round pairs in all datasets) with the ability to map ~20 proteins in the same field of view, in the same intact tissue (or cell culture) sample, through a multiscale staining strategy, finely tuned sample staining and washing steps, and optimized image registration pipelines. We showed the power of multiExR to reveal colocalization of glutamate receptor subunits with amyloid nanodomains in Alzheimer's model mice, and to visualize synaptic proteins in mouse cortex. MultiExR, despite its power in resolving large numbers of proteins with nanoscale precision, only requires ordinary chemical reagents, conventional antibodies, and classical microscopes (flowcharts for overall ExR workflow, and detailed choice guidance, are provided in Fig. 1 and Supplementary Fig. 1 respectively). A caveat of our technique and any technique that relies on antibody staining is that results depend on the specificity and sensitivity of the antibodies used. We relied on commercial vendors and publications to provide evidence of validation (see Supplementary Table 18 for more details). If antibody specificity is of particular concern to end users, such users can perform additional validation using knockout cell lines or tissues, or other validations appropriate to the scientific question at hand. Finally, it is important to note that multiExR image quality depends on antibody performance. To assist future users, we have provided a list of antibodies that yielded negligible signal with multiExR in the region imaged (Supplementary Table 14).

A key decision in the multiExR workflow is the choice of reference channel. We chose Lectin, a marker of blood vessels, as the millimeter-to-micron-scale reference channel, because blood vessels are present throughout the brain parenchyma, and exhibit unique morphologies that allow a researcher to visually locate the same field of view for imaging across rounds. However, we found blood vessels alone did not provide sufficient nanoscale feature density for fine-scale registration. Thus, we added neurofilament and/or glial-process markers, SMI and GFAP respectively, as well as a synaptic scaffolding protein (Homer) to the same reference channel, to facilitate nanoscale feature identification and mapping across rounds. Each of these markers is expected to be abundant in the brain areas we imaged. However, users are not limited to our choice of reference channel. Indeed, if multiExR is applied outside of the brain, in other tissues, users will have to use a different reference channel, which will need to be validated and optimized. We think any abundant, bright (high signal-to-noise), and heterogeneous (i.e., unique features at multiple length scales, from nano to macro) stain could work as a reference channel – including, potentially, a non-specific protein stain.

Many multiplexing technologies for protein visualization exist, such as cyclic immunofluorescence[16–22], fluorophore quenching[23], or use of DNA-antibody conjugates[24]. Some of these multiplexing technologies, including immunostaining with signal amplification by exchange reaction (ImmunoSABER)[25], multi-round immunostaining expansion microscopy (MiriEx)[19], magnified analysis of the proteome (MAP)[20], and decrowding expansion pathology (dExPath)[12] have been demonstrated to work with 4-fold expansion of biological specimens. Conversely, existing super-resolution imaging techniques, including stimulated emission depletion (STED)[26], structured illumination microscopy (SIM)[27], stochastic optical reconstruction microscopy (STORM)[28], and photoactivated localization microscopy (PALM)[29] are limited to five channels and therefore five proteins in a sample, owing to spectral overlap of fluorophores. Another super-resolution method, Exchange-PAINT[24,30], a variation of DNA points accumulation for imaging in nanoscale topography (DNA-PAINT)[31], can in theory be multiplexed to image more than four proteins in a single sample, but has not yet been demonstrated with 3D imaging in tissues. 3D DNA-PAINT imaging of organelles in cell cultures has been achieved[32,33]. However, these studies were only shown with 2–3 channel multiplexing in cell cultures.

Narayanasamy et al. demonstrated Exchange-PAINT for super-resolution imaging of synapses in tissue[34]. However, this approach was only shown with 2-dimensional images and is limited by the number of available secondary antibodies with compatible species. Another approach, multiplexed automated serial staining stochastic optical reconstruction (maS³STORM), demonstrated 3D super-resolution imaging of 16 targets in central nervous system (CNS) tissue, but requires a direct stochastic optical reconstruction microscopy (dSTORM)-capable microscope[35]. Yet another method, molecule anchorable gel-enabled nanoscale imaging of fluorescence and stimulated Raman scattering microscopy (MAGNIFIERS) achieved 8-plex 3D nanoscale imaging in a mouse brain slice, but requires a Raman microscope[36]. All of these approaches require custom DNA-conjugated

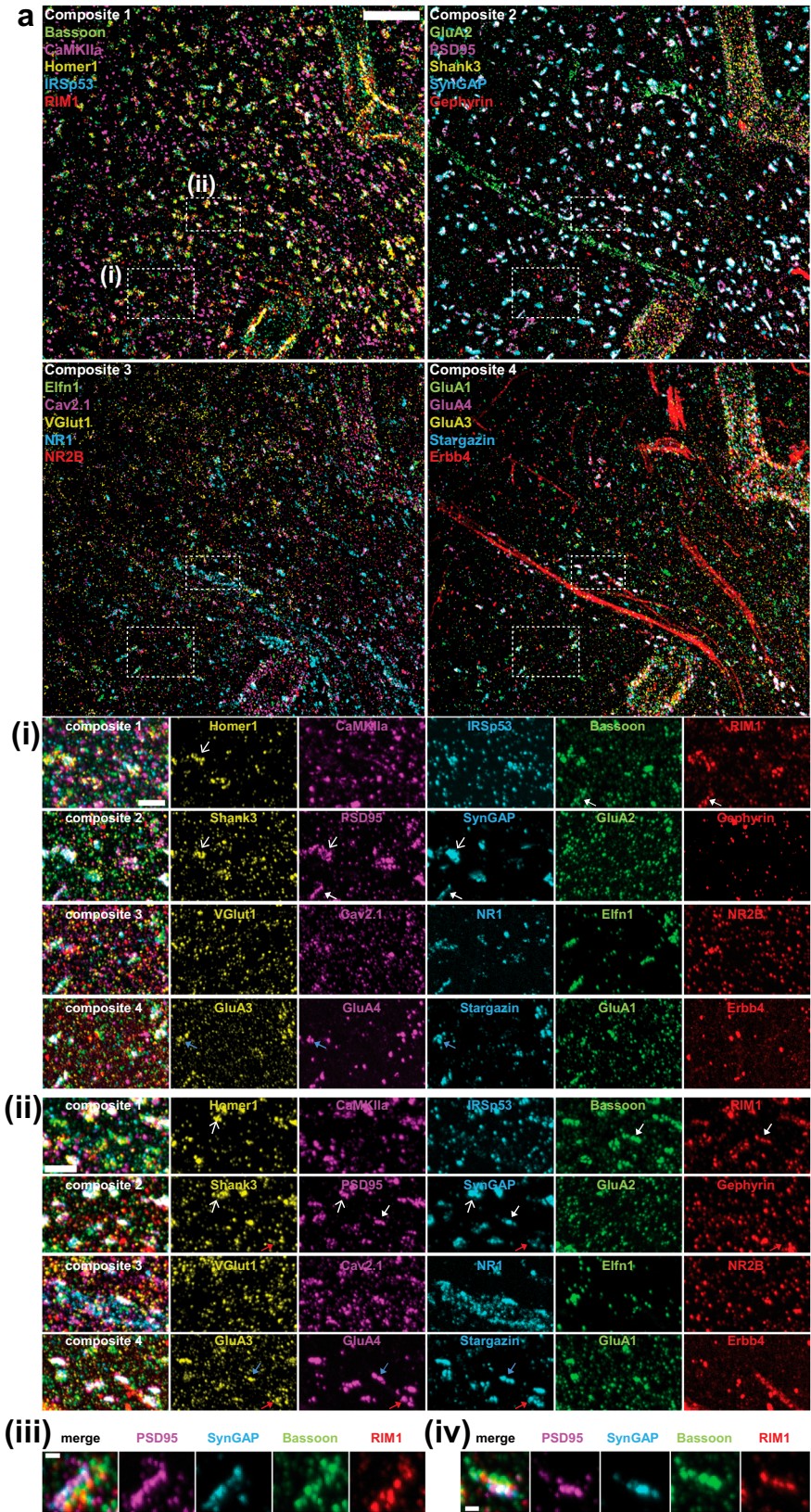

antibodies, custom imaging reagents, and/or advanced hardware such as total internal reflection fluorescence (TIRF) illumination systems, which are not readily available in most biology and neuroscience laboratories.

By comparison, multiExR can generate 3-dimensional, 20+ channel, super-resolution images of proteins in tissue sections, using widely available reagents and an inverted confocal microscope. MultiExR enables nanoscale imaging, with potentially very high multiplexing capacity, requiring only ordinary microscopes and common laboratory reagents. But the high resolution of ExR demands staining, washing, and registration strategies up to the challenge of nanoscale alignment, realized in this current paper. We note there is a speed

**Fig. 5 | 20-plex nanoscale characterization of synapses in the mouse somatosensory cortex. a** Example composite 5-channel maximum intensity projection a field of view showing synaptic proteins in mouse somatosensory cortex obtained using multiExR (from one of two mice from one batch of experiments). Scale bar, 2 μm in biological units. **i–ii** Single-channel and composite maximum intensity projections of synaptic proteins in the boxed regions from (**a**). Line-headed arrows indicate colocalized postsynaptic scaffold proteins; triangle-headed arrows indicate sandwich-like structures between pre- and postsynaptic scaffold proteins; red arrows indicate gephyrin with excitatory synaptic proteins nearby; blue arrows indicate colocalized AMPA receptors with transmembrane AMPA receptor regulatory proteins (Tarp gamma-2, Stargazin)). Scale bar, 500 nm in biological units. **iii-iv** Single-channel and composite maximum intensity projections of synaptic proteins forming sandwich-like structures from (i)-(ii). Scale bar, 100 nm in biological units.

versus measurement error tradeoff with multiExR (Supplementary Fig. 1). There is a relationship between choice of reference channel and protocol duration: dedicating more optical channels to reference stains improves registration quality at the cost of the number of target stains that can be imaged per round. It may take several weeks to acquire a 20-protein dataset with an average registration error of ~40 nm. Thus, the method may not be practical for certain applications, such as for large-sample size confirmatory experiments, especially when 4 or fewer protein targets need to be imaged. For such experiments, classical ExM or ExR may be more appropriate. As mentioned in the discussion and demonstrated in Figs. 3–5, multiExR may be most useful as a hypothesis-generating, exploratory technique, analogous to other high-resolution spatial multiplexing techniques (e.g. expansion sequencing[8]), which also require weeks for data acquisition. As with other cyclic immunofluorescence-based methods, speed bottlenecks include overnight incubations for primary and secondary antibodies and time spent imaging. If speed is a priority, users could consider shortening antibody incubation time (e.g., 2 hours at room temperature for secondary antibodies; though a shorter incubation time would need to be tested empirically for each primary and secondary antibody) or imaging a small, targeted, set of regions of interest to shorten imaging time.

The appropriateness of multiExR for an experiment depends on the required measurement scale for the underlying biological question and the distribution of registration errors achieved for a given image. That is, what is the minimum distance between puncta (for example) that one needs to measure, below which two things are considered indistinguishable, and above which things are considered separate? One useful observation is: registration error is not constant throughout an image, and thus regions can be found that enable higher precision measurements than others. For example, if 95% of the calculated registration error values fall within the range of 34-37 nm (as for one round pair in Supplementary Table 6), then a user can with 95% confidence measure distances 54-57 nm in size (taking measurement error to be registration error + resolution). We provide information on the distribution of registration errors we achieved for each field of view in each round pair in Fig. 2f, Fig. 3c, Supplementary Fig. 3b, Supplementary Fig. 4c, and Supplementary Tables 2, 4, 6, 11, and 14. The software we provide calculates 1000 estimates of registration error for every field of view (one for each randomly sampled subvolume above a signal threshold, see Methods), enabling the user to estimate the distribution of registration errors in the image. Should a reader want to examine only portions of the image that fall within a given registration error range (for example, 30 nm or less), they can crop the image to regions with registration error in this range (Supplementary Fig. 7). Given the right-skew of the distributions of registration error that we found (Fig. 2f, Fig. 3c, Supplementary Fig. 3b, Supplementary Fig. 4c), the majority of the field of view could be in an acceptable range depending on the biological question at hand.

In principle, a total protein stain such as N-hydroxysuccinimidyl(NHS)-ester bearing a fluorophore could make an excellent reference stain, as it labels densely and across scale, as is being explored for expansion-based connectomics[37]. In practice, we found that NHS-ester staining (e.g., an overnight incubation with 2 μM NHS-ester dye (Atto 647 N NHS ester, Millipore Sigma 18373) in NHS ester staining buffer (5x SSCT, 5x SSC + 0.1% Tween 20) on a shaker at RT, and washing three times in 5x SSCT for 30 min each at RT)[38]) was different enough from our antibody staining condition, that we were concerned that it would add additional complexity to the protocol beyond what it exhibits now. Given that our antibody-based reference channel strategy could achieve small registration errors, we did not pursue NHS-ester staining further, in the current study.

Alzheimer's Disease is a devastating neurodegenerative disorder marked histopathologically by synapse loss, amyloid-beta plaques, and tau neurofibrillary tangles[39,40]. 5xFAD mice were engineered to carry 5 familial Alzheimer's Disease mutations in amyloid precursor protein and presenilin, which increase the production of Aβ plaque formation, to model Alzheimer's Disease pathology in an accelerated manner[13]. We used multiExR to examine Aβ and synapse pathology in 5xFAD mouse cortex, comparing the expression and localization of 23 different proteins relative to wild-type mouse cortex. This approach, which detects the nanoscale localization of 12F4, 6E10, and D54D2 species, which identify different isoforms and conformations of Aβ, provides a simple-to-execute alternative to previous approaches to detect Aβ in synapses, such as array tomography and STORM[41]. As has been found previously[42,43], we observed dramatic synapse alteration in 5xFAD mouse cortex, marked by reduction in number and volume of several proteins. Finally, we observed colocalization of synaptic proteins and Aβ nanoclusters, in particular GluA2 and GluA4. This finding is in line with previous work showing deficits in AMPA receptor function after amyloid exposure[41,44,45] and may reflect pathological aggregation of synaptic proteins in the 5xFAD brain, similar to what we observed with Nav and Kv channels in our previous work[4]. Our comparison of 5xFAD and WT brains demonstrate the utility of multiExR in detecting and quantifying previously unappreciated nanoscale, multiprotein pathology in disease states.

The synapse is a densely packed biomolecular environment where thousands of proteins interact to facilitate rapid synaptic transmission and downstream signaling cascades. Synaptic proteins are numerous, often form complexes with one another, and are often mutated in psychiatric and neurodevelopmental disorders[46,47]. Understanding the abundance and (co)localization of these proteins at the synapse is critical to our understanding of neuronal communication in healthy and disease states. To demonstrate the utility of multiExR in visualizing synaptic proteins towards this purpose, in a second demonstration, we visualized 20 synaptic proteins in the same tissue specimen of the mouse somatosensory cortex. In the future, detailed analysis of such high-dimensional datasets could facilitate the classification of synapse types and states. Nanoscale multi-protein colocalization as revealed by multiExR could be used to generate novel hypotheses regarding protein-protein interactions.

## Methods

### Brain tissue preparation

All procedures involving animals were in accordance with the US National Institutes of Health Guide for the Care and Use of Laboratory Animals and approved by the Massachusetts Institute of Technology Committee on Animal Care. Mice housing condition is normal dark/light cycle, ambient temperature, and humidity. Both male and female wild-type (WT) (n = 3 C57BL/6, 6-8 weeks of age, from JAX), aged WT (n = 2 C57BL/6, 12-13 months of age), and 5xFAD mice (n = 2, 12-13 months of age, from the Mutant Mouse Resource and Research

Center) were deeply anesthetized using isoflurane in room air. Mice were perfused transcardially with ice-cold 15 mL of 2% (w/w) acrylamide (AA) in phosphate-buffered saline (PBS) followed by 15 mL of 23.2% (w/w) acrylamide (AA) and 2.7% (w/w) paraformaldehyde (PFA) in PBS (e.g., 15 g acrylamide added in 32.5 mL deionized water with 5 mL 10x PBS and 12.5 mL 16% PFA solution). Brains were harvested and post-fixed in the same fixative overnight at 4 °C. Fixed brains were transferred to 100 mM glycine for 6 hrs at 4 C, then transferred to PBS. The brain regions for expansion were dissected from 50-100 μm free-floating slices cut on a vibrating microtome (Leica VT1000S) in PBS. Sagittal sections were used for Figs. 3 and 4, and coronal sections were used for all others. Considering that the final expansion will be 18-20-fold, it is recommended that the starting tissue size is smaller than 3 × 3 mm for easy handling.

### Cultured neuron preparation

Cultured mouse hippocampal neurons were prepared from postnatal day 0 or 1 Swiss Webster mice (Taconic, both male and female mice) as previously described[48]. The coverglasses in the 24-well plates were pre-treated with diluted Matrigel (250 μL Matrigel in 12 mL DMEM (Dulbecco's Modified Eagle Medium)), and 160,000-200,000 cells were plated in each well. Neurons were grown for 14 d to 1 month in the incubator (37 °C and 5% CO2 in a humidified atmosphere). For fixation, neurons were briefly washed with PBS, and fixed with 4% PFA in PBS for 10 min at room temperature and 0.61% (w/w) PFA + 1% acrylamide in PBS for 6 h at 37 °C. After washing with PBS, samples were stored at 4 °C before expansion.

### Expansion of brain tissue slices and cultured neurons

The first gelling solution was prepared (7.4% (w/w) sodium acrylate (SA), 2.5% (w/w) AA, 0.08% (w/w) N,N'-methylenebisacrylamide (Bis), 0.2% (w/w) ammonium persulfate (APS) initiator, 0.16% (w/w) tetramethylethylenediamine (TEMED) accelerator, and 0.01% (w/w) 4-Hydroxy-TEMPO (4HT)) and 2M NaCl in 1x PBS base without adding APS. The dissected brain slice or cultured neuron plated coverglass was placed between two #1.5 coverslips separated by two pieces of #1.5 coverslips. After vortexing the gelling solution with APS, the excess gelling solution was added around the tissue. After incubation at 4 °C for 30 min, the sample was incubated at 37 °C for 30 min to 2 hrs until gel formation. The tissue or neuron-containing gel was obtained by trimming excess gel. It is helpful to cut the gel into an asymmetric shape to maintain tissue orientation. The gel was incubated in denaturation buffer (200 mM sodium dodecyl sulfate (SDS), 200 mM NaCl, and 50 mM Tris pH 8) for 1 h at 95 °C using a thermocycler. The denatured gel was placed in 6-well plates and washed at least twice using deionized water (DIW) for 15 min each. For the re-embedding step, expanded 1st gel was placed in 6 well plate and incubated in the re-embedding solution (13.75% (w/w) AA, 0.037% (w/w) Bis, 0.023% (w/w) APS, 0.02% (w/w) TEMED) twice for 1 h each on the shaker at room temperature. The first re-embedding solution was replaced with a freshly made second re-embedding solution. The gel was transferred and placed between coverslips, avoiding bubble formation. The gel chambers were placed in a plastic zipper storage bag for 5 min nitrogen purging and incubated for 1–2 hrs at 45 °C. For the 3rd gelling step, the re-embedded gel was placed in a 6 well plate and incubated in the 3rd gelling solution (7.4% (w/w) SA, 2.5% (w/w) AA, 0.04% (w/w) Bis, 0.023% (w/w) APS, and 0.02% (w/w) TEMED and 2M NaCl in 1x PBS) twice for 1 h each on the shaker at room temperature. The gel was transferred and placed between coverslips with the additional gelling solution to avoid bubbles. The gel chambers were placed in a plastic zipper storage bag for 5 min nitrogen purging and incubated for 1 h at 60 °C. The gel was transferred to a petri-dish with DIW. The gel was fully expanded in DIW by overnight incubation and changing excess water 2–3 times per 2 hrs on the following day. The gel was trimmed axially from the tissue or neuron-containing portion at the bottom to reduce the thickness to 1 mm to facilitate subsequent immunostaining and imaging. We opted to further stabilize the trimmed gel for using additional re-embedding step with reduced AA concentration (2% (w/w) AA, 0.037% (w/w) Bis, 0.023% (w/w) APS, 0.02% (w/w) TEMED) so as to maintain gel integrity during multiple rounds staining, stripping, and imaging.

### Immunostaining of expanded tissues

The gel was incubated in blocking buffer (0.5% Triton X-100, 5% normal donkey serum (NDS) in PBS) for 1–2 hrs at room temperature. The gel was incubated with primary antibodies in staining buffer (0.25% Triton X-100, 5% NDS in PBS) overnight at 4 °C. The gel was washed in 1x PBST washing buffer (0.1% Triton X-100 in 1x PBS) 6 times for 30 min each on a shaker at room temperature. The gel was incubated with secondary antibodies in staining buffer overnight at 4 °C, and washed in 0.05x PBST washing buffer (0.1% Triton X-100 in 0.05x PBS) 6 times for 30 min each on a shaker at room temperature. The antibodies against three target proteins per round were stained with 488, 546, and 633 nm channels, and the Lycopersicon Esculentum Lectin with 594 nm channel (Vector Laboratories, DL-1171-1) was co-stained every round to serve as a reference channel. The images of the multi-ExR sample were obtained using a Nikon CSU-W1 or SORA confocal microscope with 100% laser power and 1 s exposure time per channel. The global tiled image using 4x and 10x objectives and Element software with annotation of ROI marking were used to find the same field of view over multiple rounds imaging. The same field of view is located and re-imaged through the following process. In the first round of staining and imaging, we obtain a large (~500um x 500um, covering most of the gel) mosaic image of the reference channel, followed by an image of a smaller region of interest, both with the 10x objective. These mosaic images of the reference channel are used to guide the experimentalist in finding the same field at increasing magnification in later rounds using the same reference channel. The 40x water immersion objective was used to collect the fields of view shown in the manuscript with a 0.25 μm z-step size. The gel was placed on the glass bottom 6 well plate, covered with plate-sealing film for 30 min before starting the imaging session to stabilize the gel to prevent drifting during image collection. For the stripping of antibodies, the gel was incubated in denaturation buffer with 100 mM beta-mercaptoethanol for 1 h at 95 °C and washed 4 times with excess PBS. The next round of immunostaining was repeated by blocking, staining, imaging, and stripping the gel as described.

### Expansion factor measurement

The 20x expansion factor for ExR has been demonstrated previously[4]. A second re-embedding step was added after the final expansion to strengthen the gel for multi-round gel handling. To determine the effect of second re-embedding on the expansion factor, we measured gel size after the second re-embedding. Before the second re-embedding step, six gels were excised to the height of 1.5 cm. For the second re-embedding step, expanded 3rd gels were incubated in re-embedding solution (2% (w/w) AA, 0.037% (w/w) Bis, 0.023% (w/w) APS, 0.02% (w/w) TEMED) twice, replacing the first solution with freshly made re-embedding solution for 1 h each time on a shaker at room temperature. The second re-embedded gels were washed in 0.05x PBST washing buffer (0.1% Triton X-100 in 0.05x PBS) 4 times for 1 h each on a shaker at room temperature, then the size of gels were measured, and decreased by 10% on average, leading to a final expansion factor of 18x.

### Image preprocessing and registration

All custom image processing and analysis scripts for all analyses in this manuscript are available at https://github.com/schroeme/multi-ExR. First, background was subtracted from image stacks using ImageJ/Fiji's Rolling Ball algorithm with a radius of 50 pixels. After background

subtraction, images from later rounds were registered in x, y, and z space to the first imaging round (Fig. 1c). One or more channel(s), always including the reference channel (whether single or three-stain), were designated to serve as the reference channel between subsequent rounds. We created a toolbox for both global and local alignment of multiExR imaging rounds, consisting of (with reference to Supplementary Fig. 1: **1(i)** a 3D scale-intensity feature transform (SIFT)-based global registration algorithm previously used for multiround RNA multiplexed-imaging alignment (ExSeqProcessing registration[8,9]), **1(ii)** an alternative intensity-based global registration algorithm (Elastix[10]), and **1(iii)** a point-based alignment step for refinement of local structures such as synapses (Fig. 1c).

(1) Feature-based global registration. The first is a previously-described algorithm utilizing 3D scale-invariant feature transform (SIFT) for keypoint detection and a 3D affine transform[8,9], available at https://github.com/dgoodwin208/ExSeqProcessing. Briefly, keypoints are detected, features are constructed, subsequently matched between image volumes. These matched features are used to calculate a warp for one image into the space of another. For all datasets, the first imaging round was used as the reference round for registration. We used the following parameters from the publicly available ExSeqProcessing repository registration pipeline: downsample_rate = 4 and pyramid_scale = [1:9]. For the staining rounds of all fields of view from the validation dataset (Fig. 2), the four fluorescence channels were normalized and summed to serve as the reference channel, per the default ExSeqProcessing configuration. For the stripping rounds of the primary validation dataset, all rounds of the secondary validation dataset, and all rounds from the synaptic dataset, only the multi-protein reference channel was used to detect features and calculate the warp. For the 5xFAD dataset, all channels were used as reference channels, but they were not normalized and summed prior to feature detection, as this was found empirically to improve registration quality. For difficult registrations, we recommend trying a variety of registration configurations, including using one or multiple channels as reference, with and without normalization, as described in Supplementary Fig. 1. The ExSeqProcessing pipeline was run using cuda = True on a GeForce RTX 2080 Ti GPU, allowing multiple fields of view to be registered in a few hours.

(2) Intensity-based global registration. Expansion microscopy images of the brain often contain neural structures with various scales, which is challenging to align well simultaneously. Thus, for empirically difficult (i.e., noisy or highly shifted between rounds) fields of view that could not be registered using the ExSeqProcessing pipeline, we implemented an alternative intensity-based method for coarse structures, available at https://github.com/dongliaw/ExM-Toolbox/tree/ck/mExR. We first pre-processed the image volumes to remove fine structures and image noise with the Non-Local Means denoising method (Supplementary Note 1) and then the adaptive thresholding method (Supplementary Note 2) to mask them out. Then, we used the Elastix[10] package to estimate the global affine transformation by minimizing the mutual information of the intensity of matched voxels. Finally, we use the first-order B-Spline Interpolator with the estimated transformation to compute the spatial mapping. We used this registration method for four rounds in one field of view in the secondary validation dataset (Supplementary Fig. 3), for one round in one field of view in the synaptic dataset (Fig. 5), for several fields of view obtained in earlier optimization experiments that were not included in this manuscript, and for some other fields of view in this to confirm its utility (Supplementary Fig. 6).

(3) Point-based local registration. After the global alignment of coarse structures, we aimed to improve the registration of fine-scale structures of interest, e.g., synapses. We first extracted synapses by removing image noise with Non-Local Means denoising, followed by the adaptive thresholding method to mask both the noise and the coarse structures. Then, we computed the centroid of synapses to obtain a point cloud for each volume. Next, we applied the Iterative Closest Point (ICP) algorithm (Supplementary Note 3) to find point matches and compute affine transformation iteratively. Finally, we used the radial basis function (RBF) for interpolation to generate a dense deformation field from the sparse matches[11] (Supplementary Note 4). The dense deformation field thus produced is used for interpolating pixels in the coordinate space, which yields the warped volume. A tutorial for this method is provided at: https://github.com/dongliaw/ExM-Toolbox/blob/ck/mExR/tutorial.ipynb.

For experiments in this manuscript, we mostly used the feature-based global alignment method (**3(i)** in Supplementary Fig. 1c, ExSeqProcessing registration), which produced registered images in an acceptable error range for the demonstration of our technology. However, during development, we noticed that the intensity-based global alignment (**2(ii)** in Supplementary Fig. 1c) sometimes works well on failure cases in the feature-based method (Supplementary Fig. 6a-c). In addition, the point-based local alignment step (**2(iii)** in Supplementary Fig. 1c) can improve synapse alignment (Supplementary Fig. 6d). Thus, we provide all methods here as a toolbox, from which users can choose the best-performing set of algorithms for their data.

## Quantification of registration accuracy

To avoid image stack edges that were empty for some imaging rounds after registration (because warping often involved translation in the z-axis), image volumes were cropped to a 61-slice stack of mutually overlapping volume for each round. Registration accuracy was quantified based on a previously described method[8]. For the primary validation dataset, we chose to quantify registration error using the SynGAP channel, which had higher SNR than the bassoon channel (Fig. 2). For the secondary validation dataset, we quantified registration error using the 4-stain reference channel, to be consistent with later experiments where only the reference channel is available to calculate registration error. For the synaptic and 5xFAD datasets, we used the multi-channel reference channel to quantify registration error. Volumes were converted to grayscale and cropped to slices with at least one nonzero pixel. Then, we calculated a normalized cross-correlation of 1,000 subvolumes (each of size $100 \times 100 \times 61$ pixels), randomly chosen across the imaged field of view, excluding the edges ($18.5 \times 18.5 \times 0.85$ microns in size, in biological units). All subvolumes analyzed had greater than 1% of voxels above an intensity threshold of the 99th percentile intensity of the whole field of view. The registration error was calculated as the mean of the offsets in maximum normalized cross-correlation between each pair of rounds for each subvolume in each dimension. Violin plots of estimated population density were created in GraphPad Prism, and outliers were removed using the ROUT method (Q = 1%).

## Calculation of feature density in the reference channel

This analysis was performed on unregistered images. A binary mask of each channel was created as follows: conversion to grayscale (min-max normalization using MATLAB's "mat2gray"), binarization using a threshold at the 99.5th percentile value of the volume intensity distribution, and removal of connected components smaller than $50 \times 50 \times 50$ nm³ in size. The reference channel mask was taken as the union (using MATLAB's "or" function) of all individual channels. We then identified 3D-connected components from the binary stack using

MATLAB's "bwconncomp" function, with a pixel connectivity of 26, meaning that pixels are connected if their faces, edges, or corners touch. Fraction of volume occupied by the reference channel was calculated as the number of nonzero pixels in the mask divided by the total number of pixels in the volume. Minimum (maximum) feature size percentage was calculated as the number of nonzero pixels in the smallest (largest) connected component (Supplementary Table 1).

### Analysis of primary 7-round validation dataset

To avoid image stack edges that were empty for some imaging rounds after registration (because warping often involved translation in the z-axis), image volumes were cropped to a 61-slice stack of mutually overlapping volume for each round. To quantify the stripping and restaining efficiency of the validation dataset, in which the SynGAP and Bassoon were repeatedly stripped and stained over seven rounds using the same experimental conditions and microscope settings (Fig. 2), we automatically identified and counted putative synapses. The binary image was created as follows: conversion to grayscale, binarization using a threshold at the 99.5th percentile intensity value, median filtering with a radius of 9x9x3 pixels, subtraction of the Lectin channel mask, morphological closing using a disk structuring element of radius 250 nm, size filtration with a lower limit of 50x50x50 nm$^3$. The mask of the Lectin channel was created as follows: conversion to grayscale, binarization using a threshold at the 99.5th percentile intensity value, median filtering with a radius of 9x9x3 pixels, and morphological closing using a disk structuring element of radius 2 μm. We then identified 3D connected components from the filtered binary stack using MATLAB's "bwconncomp" function, with a pixel connectivity of 26, meaning that pixels are connected if their faces, edges, or corners touch. Objects (putative synapses) were defined as 3D connected components of the filtered, binary image volume (Fig. 2c).

In order to determine whether nanoscale synaptic properties were maintained over seven rounds of stripping and staining, we quantified the number of puncta, mean puncta volume, and brightness (as measured by absolute intensity and SNR) of manually-identified synaptic ROIs (Fig. 2d, e and Supplementary Fig. 2b, c). Two-dimensional ROI boundaries were selected in Fiji using the rectangle tool, based on the presence of both Bassoon and SynGAP staining in the first round. The ROI was cropped in 2 dimensions using the x- and y- boundaries from Fiji's ROI manager, with the z-boundary extending 15 slices in each direction from the center plus one frame. Synaptic ROIs were processed as follows: conversion to grayscale, binarization using a threshold at the 99.5th percentile intensity value, and size filtration to remove puncta less than 20x20x20 nm$^3$, which are likely noise. Synaptic protein puncta were defined as 3D connected components of the filtered, binary image volume (pixel connectivity of 26). Puncta volume was calculated from the binary mask volume using MATLAB's "regionprops3" function, multiplied by an average voxel size conversion factor of 1.2073 ×10$^{-6}$ um$^3$ per voxel (weighted average of x, y, and z spatial sampling in post-expansion units, cubed). SNR was calculated as the mean intensity in the masked region (within synaptic protein puncta) divided by the mean intensity in the inverse of the masked region (within the background). Mean absolute intensity was calculated as the mean intensity of pixels within the masked region.

### Quantification of 5xFAD vs. WT datasets

To avoid image stack edges that were empty for some imaging rounds after registration (because warping often involved translation in the z-axis), image volumes were cropped to a 61-slice stack of mutually overlapping volume for each round. To quantify the volume of Aβ species (Fig. 3d), image volumes for each of the Aβ channels were processed as follows: binarization using an absolute intensity threshold (Supplementary Table 15), determined based on 5 standard deviations above the mean intensity of the 5xFAD fields of view, 3D

median filtering with radius 5x5x3 voxels (in x, y, and z, respectively), subtraction of the reference channel, to avoid quantification of non-specific staining along blood vessels, and exclusion of small objects (likely noise) under 100 voxels in volume. The mask of the reference channel was created as follows: binarization using a threshold at the 99th percentile intensity value, median filtering with a radius of 5x5x3 pixels, and morphological closing using a disk structuring element of radius 10 pixels. Total volume was calculated as the number of nonzero pixels in the binarized image volume, multiplied by an average voxel size conversion factor of 1.2073 ×10$^{-6}$ um$^3$ per voxel (cube of weighted average of x, y, and z spatial sampling rates).

To quantify differences in synaptic protein expression between 5xFAD and WT mice (Fig. 3e), synaptic channel image volumes were processed as follows: binarization using an absolute threshold (Supplementary Table 16), determined based on 5 standard deviations above the mean intensity of the dimmest WT field of view, 3D median filtering with radius 5x5x3 voxels (in x, y, and z, respectively), subtraction of the reference channel, to avoid quantification of non-specific staining along blood vessels, size filtration to include only puncta with volume greater than 100 voxels and less than 5000 voxels. The mask of the reference channel was created as described above, but with a disk structuring element of radius 10 pixels. The number of objects and total volume in each synaptic channel were calculated as described above. For quantifying effect size, we used a linear mixed effects model (Python's statsmodels[49] "mixed_lm") to avoid type I error due to pseudoreplication. The model was set up as follows: model = smf.mixedlm("Vol ~ Group", data, groups=data["Animal"]), where "Animal" indicates animal ID, and "Group" assignment was either WT or 5xFAD. Jupyter notebooks with relevant code are available at https://github.com/schroeme/multi-ExR.

We analyzed colocalization of Aβ species and synaptic proteins in 5xFAD brains within manually-identified ROIs containing Aβ nanoclusters (Fig. 4). The ROI was cropped in 2 dimensions using the x- and y- boundaries from Fiji's ROI manager, with the z-boundary extending 18 slices in each direction from the center plus one frame. The volume and number of puncta of each protein were calculated as described above, after binarization using a threshold of 4 standard deviations of the mean intensity within the nanocluster ROI, 3D median filtering with radius 5x5x3 voxels, and size filtration with a minimum volume of 20 voxels. The fraction of volume mutually overlapped with D54D2 for GluA1-4 (Fig. 4c) was calculated as the number of nonzero pixels in the intersection between GluA1-4 and D54D2 binary masks, divided by the number of nonzero pixels in the D54D2 binary mask. For this analysis, a size filter of 20 voxels was also used. For overlap colocalization analysis (Fig.4c), we excluded ROIs with visible offset between the Aβ channels, representing residual registration error, and kept 44/71 ROIs from 8/9 fields of view.

### Choice of median and size filters for nanoscale synaptic and Aβ puncta

The reader will note that the median and size filters used in the analyses described above are varied, depending on the biological goal at hand. Our selection of size filter for each of these analyses was based on empirical findings related to what produced a reasonable mask to exclude high spatial frequency noise based on visual inspection, as well as the expected size of a synapse (e.g., the synaptic cleft is about ~20 nm). Too strict a size filter (e.g., requiring puncta to be >150 voxels or 50-60 nm) leads to exclusion of puncta within actual synapses, while too permissive a size filter (e.g., <1 voxel or <10 nm) may be ineffective at removing high spatial frequency noise. We acknowledge that thresholding followed by filtration-based methods of puncta/synapse identification are, due to inevitable variability, bound to produce some false positives and false negatives. This is why we used manually-identified ROIs for detailed analyses within putative synapses or beta-amyloid nanoclusters, and decreased the size filter used on manually-

selected ROIs to reduce the chance of excluding actual puncta in such regions, which we are confident contain the biological structures of interest. To investigate the effect of our choice of size filter on our results, we conducted four parameter scans as detailed below, the results of which are summarized in Supplementary Fig. 8.

1. We calculated signal-to-noise ratio in the validation dataset (Supplementary Fig. 2b) with various size filters in the staining rounds. The size filters tested were chosen to span the range from 0 (no size filter) to 191 voxels ($60 \times 60 \times 60 nm^3$; Supplementary Fig. 8a)

2. We calculated total puncta volume of synaptic proteins within manually-identified beta-amyloid nanoclusters (Fig. 4b) with various size filters, chosen to span the range from 0 to 100 voxels (Supplementary Fig. 8b). We chose to proceed with a filter of 20 voxels, because this was the largest filter for which no amyloid-beta puncta, the basis of which these ROIs were manually identified, were excluded based on size.

3. We calculated the fraction of D54D2 volume occupied by GluA2 (Fig. 4c), with various size filters applied to both channels, ranging from 0 (no filter, which is what we report in the manuscript) to 180 voxels (over double what was used for the analysis in Fig. 4b, see Supplementary Fig. 8c). We chose a size filter of 20 voxels to be consistent with the analysis in Fig. 4b.

4. We calculated the fraction of D54D2 volume occupied by GluA1-4 (Fig. 4c) both with and without a median filter of size 5x5x3 voxels, using a size filter of 20 voxels (Supplementary Fig. 8d). Eliminating the median filter did not affect the pattern of which AMPAR subunits had the most colocalization with D54D2, but did increase the numerical value of the fraction of D54D2 volume containing GluA1-4, especially in the case of GluA3. With no median filter, there were still more ROIs for which there was no GluA4 contained within a D54D2 punctum than for GluA2 (0% for GluA2 vs. 22.7% for GluA4; Chi-square = 11.28, p = 0.0008, $n$ = 44 nanocluster ROIs from 8 fields of view from 2 5xFAD mice). Furthermore, with no median filter, conclusions based on the linear regressions shown in Fig. 4d and Fig. 4f are unchanged. We chose to proceed with a median filter to be consistent with other analyses in the paper.

The results from these tests suggest that within a certain range, the choice of size or median filter does not greatly affect our results or conclusions, but is a choice that each user should tailor as appropriate for their biological question.

### Reagents
Lists of reagents used in this study, as well as the composition of gelling and denaturation solutions, are provided in Supplementary Tables 17-19.

### Statistics and reproducibility
No statistical method was used to predetermine sample size for this study. In some experiments, channels with poor staining quality (very low detectable signal) or that were imaged in an earlier staining round were excluded, as detailed in Supplementary Tables 5 and 10. Rationale and methods for excluding data during analysis (e.g., ROIs with high registration offset and outliers in registration error quantification) are described above. The experiments were not randomized, nor were experimenters blinded to genotype during data acquisition or analysis.

### Reporting summary
Further information on research design is available in the Nature Portfolio Reporting Summary linked to this article.

### Data availability
Preprocessed, registered data are available for download from Harvard Dataverse at https://doi.org/10.7910/DVN/JJBULY. Processed data derivatives used to generate plots (i.e. Source Data, in Excel format) are available for download with this paper. Blank cells in the Source Data files are from outliers removed as described in the Methods section. Source data are provided with this paper.

### Code availability
All custom image processing and analysis scripts for all analyses in this manuscript are available at https://github.com/schroeme/multi-ExR (v0.1, https://zenodo.org/records/13646611), https://github.com/dgoodwin208/ExSeqProcessing, and https://github.com/donglaiw/ExM-Toolbox/tree/ck/mExR (https://zenodo.org/records/13750923).

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

## Acknowledgements

We thank Daniel Goodwin, Yosuke Bando, and Atsushi Kajita for their help adapting the ExSeqProcessing registration pipeline for multiExR. We thank Dr. Li-Huei Tsai for the gift of 5xFAD mice and feedback on the manuscript. G.F. and J.K. were supported by the Tan-Yang Center for Autism Research at MIT. M.E.S was supported by the MathWorks Science Fellowship, the Collamore-Rogers Fellowship, the National Science Foundation Graduate Research Fellowship under Grant No. 1745302, and NIH 1F31MH133329-01. D.W. is supported by NSF IIS-2239688. E.N. is supported by the Alana Foundation USA, Halis Family Foundation, Lester A. Gimpelson, Donald and Glenda Mattes, David B. Emmes, and Thomas A. Stocky and Avni U. Shah. E.S.B. was supported by Tom Stocky, NIH 1R01EB024261, Kathleen Octavio, Lore McGovern, Good Ventures/Open Philanthopy, Lisa Yang, NIH 1R01AG070831, HHMI, the European Union's Horizon 2020 research and innovation programme (grant agreement No 835102), NIH 1R56AG069192, NIH R01MH124606, NIH R37MH080046, and John Doerr. The funders had no role in study design, data collection and analysis, decision to publish, or preparation of the manuscript.

## Author contributions

J.K. contributed key ideas, designed and performed experiments and interpreted data for all projects, and wrote and edited the manuscript. M.E.S. contributed key ideas, designed experiments, designed and implemented analysis, visualization, and statistical tests for all projects, and wrote and edited the manuscript. E.S.B. supervised the project, initiated work, contributed key ideas, designed experiments, helped with data analysis and interpretation, and wrote and edited the manuscript. Y.L., E.Y., and K.T. performed data collection and wrote the methods. C.K. implemented the point-based registration method with extensive result analysis and wrote its description. T.B.T. aided with the design of nanoscale imaging of synaptic proteins and edited the manuscript. M.Z. provided mouse brain tissues and antibodies. D.P. provided cultured neurons. E.N. edited the 5xFAD-related parts in the manuscript. D.W. adapted the intensity-based registration method, supervised the development of the point-based registration method, and edited the registration method description. G.F. provided supervision and edited the manuscript.

## Competing interests

J.K., M.E.S., and E.S.B. are co-inventors on a patent application for multiExR. The remaining authors declare no competing interests.
