## [Peer Review File · Nature Communications]

Reviewers' Comments:

Reviewer #1:

Remarks to the Author:

The authors present a multiplexing strategy for expansion microscopy including an antibody stripping method to enable iterative staining in expanded brain tissues and a registration method to reconstruct the multiplexed image. As proof-of-concept, the authors showcased the multiplexing of 20 proteins at 10-40nm resolution in wildtype and AD mouse brain tissues. As the authors point out, this work is incremental from their previous report of Expansion Revealing. The text is largely technical and not easy to read. The imaging and findings also lack sufficient validation. I am not convinced why multiExR is essential for the observation of AMPAR/Amyloid colocalization. I am further unconvinced that this method, in its current form, will be easily implemented outside Boyden lab. Competing nanoscale multiplexing methods have recently emerged in other labs' preprints. In comparison, the manuscript here seems unpolished and hastily put together.

Specific comments include:

- 1) The authors should present evidence that the 20 antibodies are validated and specific.
- 2) 'a reference channel with only nanoscale features is difficult to align at the macroscale'. What does this mean? Why is it difficult?
- 3) What is 'adequate feature density'?
- 4) Why is there not a neuronal marker to visualize the neuronal dendrites and axons that make these synaptic contacts?
- 5) Fig. 2 Round 1 and Round 2 show visible differences in the ratio between SynGAP and Bassoon staining. If this is an artifact of stripping, it needs to be addressed.
- 6) Which brain region is shown in Fig.3? How is the brain sliced? Some basic information should be at least mentioned in the main text/figure caption to help understand the experiment.
- 7) Why does the observation of AMPAR aggregation with amyloid require nanoscale imaging? The authors may want to present evidence that this cannot be seen by diffraction-limited confocal microscopy of non-expanded samples.
- 8) It is difficult to evaluate the (co)staining of different synaptic proteins in Fig.3&4&5. In particular, multiple proteins within the same field of view share the same pseudo-colors, which is confusing! The authors may want to show a panel of individual synapses, each with individual channels of synaptic proteins.
- 9) The vGluT1 signal in Fig.5(ii) does not seem to colocalize with any of the synaptic markers. One would also expect that the vGluT would fill up much of the presynaptic bouton around the Bassoon signal. Instead, vGluT looks like scattered puncta in the image.

Reviewer #2:

Remarks to the Author:

The authors reported a super resolution imaging method named multiplexed expansion revealing (multiExR), which enables high-fidelity visualization of >20 proteins in the same specimen, through serial rounds of staining and imaging. multiExR exhibits round-to-round registration error as low as 10-40 nm. The authors precisely mapped 23 proteins in the brain of 5xFAD Alzheimer's model mice, and found reductions in synaptic protein cluster volume, as well as co-localization of specific AMPA receptor subunits with amyloid-beta nanoclusters. They also visualized 20 synaptic proteins in specimens of mouse primary somatosensory cortex. This work is a good example of how expansion-based microscopy could contribute to the pathological analysis in practical applications. The authors provided quite sufficient data to support their idea. I only have a few suggestions.

1. There are too many parentheses in the Results section, making it hard to read. Please rearrange those sentences.
2. In the Main text section, adding some more introduction on expansion microscopy would be helpful for readers who are not that familiar with this field.
3. Did the stripping process affect the reference channel? Please add more explanation on this issue.
4. In the Discussion section, the authors said "map many proteins at once", "map ~ 20 proteins at once". This is inappropriate because multiplexing is realized in a sequential way.
5. In the Discussion section, the comparison of multiExR with other super resolution imaging methods need more thorough discussion, especially for the comparison with Exchange-PAINT. The statement "but requires specialized reagents and equipment" is not convincing.
6. Line 460, please add more explanation on how to realize that "the same field of view is imaged again".
7. Figure 3 and Figure 5, in the enlarged images, scale bars are missing.
8. Figure 4, in the merged images, there seems to be obvious displacement between the blue channel and others. Please add some discussion on this issue.

The manuscript can be accepted if the above issues have been properly dealt with.

Reviewer #3:

Remarks to the Author:

Kang et al. introduced a method named multiplexed expansion revealing (multiExR),

which achieved a resolution of approximately 20nm and the ability to detect roughly 20 proteins in a single sample. The team successfully mapped 20 synaptic proteins in a sample from the mouse primary somatosensory cortex. Notably, the study illustrated the co-localization of GluA2 and GluA4 glutamate receptor subunits with amyloid-beta nanoclusters in Alzheimer's model mice. This accessible and user-friendly method employs commonly available reagents and standard microscopes, indicating significant promise for future applications, especially with rare and valuable samples.

Two notable innovations in the study are worth highlighting: 1) the method's ability to sustain reliable signals even after up to 7 rounds of stripping and re-staining, and 2) the implementation of a 3D scale-intensity feature transform-based global registration algorithm enabling both global and local alignment of multiExR imaging. These two features enabled multiplexed super-resolution imaging of brain tissue using standard staining reagents and microscopes. Consequently, the following concerns primarily revolve around these two points:

1. Insufficient stripping

This method made use of beta-mercaptoethanol-containing denaturation buffer to break the antibody disulfide bonds. It is unclear if all residual antibodies are effectively removed or if they might impact subsequent rounds of staining or epitope accessibility through accumulation.

1). Figure 2a raises questions about the persistence of Bassoon signal within lectin after round 1 stripping, in contrast to its absence after round 2 stripping. Would there be a possibility that there might be insufficient stripping for larger structures?

2). In Figure 2c, the number of Bassoon detected in round 1 is lower than round 2 but this is not reflected in the representative images shown in Figure 2a and 2b. The discrepancy needs further clarification.

2. Backgrounds caused by multiple rounds of stripping

1) When residual antibodies accumulate over rounds of staining and stripping, it would be expected to observe a slight decrease in signal as the antibodies might block access for the next round of staining. This can be observed in Figure 2a. However, in Extended Data Figure 2d, the signal in the reference channel appears to be even higher in rounds 5, 6, and 8. More notably, the shape of the structure in the reference channel is not very consistent, particularly after rounds 4 and 9 of stripping. This inconsistency is not in line with the registration error shown in Extended Data Figure 2c.

2) In the analysis of signal-to-noise ratio, is the same threshold applied to every round of staining?

3) Is there a recommended sequence for different proteins to be stained and stripped, like most abundant or less abundant protein first stained?

3. Registration.

1) It is not clear to me why sometimes they can achieve 30nm mean registration error but later only 70nm or 100nm. Can the author list out the exact conditions when different level of registration accuracy can be achieved. As 50-100nm mean registration

error is still quite significant considering the desired resolution of ExR is 20nm. Is this caused by nonlinear distortion due to stripping and re-staining, the quality of the reference channel after multiple rounds of restaining, or other computational limitation?

2) What are the criteria of the protein to be chosen as reference channel?

Minor Concerns:

1. It is essential to acknowledge that the quality of imaging is heavily reliant on antibody performance. In the interest of user convenience and the future application of this method in the broader field, can the authors please list the antibodies and corresponding conditions that were attempted but did not yield expected results?

2. Extended Data Figure 1c lacks clarity in terms of what (A), (B), and (C) refer to.

3. The arrangement of Extended Data Figure 2 is confusing, as 2a and 2b share the same image dataset with Figure 2, whereas Extended Data Figure 2c and 2d belong to a separate image dataset, making it difficult for readers to follow.

4. Figure 3a, 3b, and 5a suffer from low image resolution, making it challenging to discern between signal and noise.

We thank the reviewers for their constructive comments. We have done additional work to address them in full, and hope the paper is now acceptable at Nature Communications.

Reviewer #1

The authors present a multiplexing strategy for expansion microscopy including an antibody stripping method to enable iterative staining in expanded brain tissues and a registration method to reconstruct the multiplexed image. As proof-of-concept, the authors showcased the multiplexing of 20 proteins at 10-40nm resolution in wildtype and AD mouse brain tissues. As the authors point out, this work is incremental from their previous report of Expansion Revealing. The text is largely technical and not easy to read.

While this work does indeed build firmly on top of the work we did earlier, the capabilities opened up by the new work are quite powerful. Simply put, our method is, we think, the most practical method of multiplexed nanoimaging that has been demonstrated on intact tissues, as detailed below. Our paper is a technology paper, and therefore technical language is required to provide readers with the necessary details required to use the tool themselves. We have made additional revisions to the text to improve its readability, per the concrete suggestions from Reviewer #2. We of course would be open to further guidance from the editor, regarding the ease of reading the text.

The imaging and findings also lack sufficient validation. I am not convinced why multiExR is essential for the observation of AMPAR/Amyloid colocalization. I am further unconvinced that this method, in its current form, will be easily implemented outside Boyden lab. Competing nanoscale multiplexing methods have recently emerged in other labs' preprints. In comparison, the manuscript here seems unpolished and hastily put together.

We have addressed the reviewer's concerns about validation (see response to question 1, below), and the necessity of multiExR for visualization of AMPAR/amyloid colocalization (see response to question 7, below). In summary, regarding validation, we used commercial antibodies that had at least one and often 2 or more of the following: knockout or knockdown validation in tissue or cell lines, high user rating on vendor websites, multiple references for publications using the antibody, and vendor-provided images of Western blots and/or immunohistochemistry performed with the antibody. We have added this information to the primary antibodies table (**Table 2**, found at the end of the Methods). Second, regarding the necessity of multiExR for observing AMPAR/amyloid cocolocalization – without ExR, the amyloid nanoclusters are hardly visible, due to their nanoscale size, and their crowded nature, which precludes staining (this was thoroughly documented, with multiple antibodies and

controls, in our previously published paper <https://www.nature.com/articles/s41551-022-00912-3>). Furthermore, discovering which proteins co-localize with these amyloid nanoclusters would be daunting without some kind of high-throughput method (we screened over 20 different proteins for co-localization with amyloid, to find the striking AMPAR result we obtained). Thus, it would have been very difficult, if not impossible, to discover this AMPAR/amyloid colocalization using earlier methods.

Regarding the method being easily implemented outside the Boyden lab: as described in the main text and methods, multiExR requires only conventional laboratory reagents, commercially available antibodies, and an inverted confocal microscope, all of which are accessible to most laboratories. The expansion step is a previously published form of iterative expansion¹ from our group; iterative expansion variants are in use by many groups, e.g. by groups at Yale², U. Geneva³, and elsewhere. Furthermore, the staining, imaging, and stripping steps are optimized forms of widely-used free-floating serial immunohistochemistry protocols (e.g., in⁴ and other references). Thus, each step of our protocol is already, in some form, in use already; what is novel, is the optimization of each the steps of our workflow for the problem at hand – multiplexed nanoimaging – and their optimized combination. Compared to other super-resolution multiplexed imaging methods, such as Exchange-PAINT^{5,6}, multiplexed automated serial staining stochastic optical reconstruction microscopy⁷, and stimulated Raman scattering microscopy-based methods⁸, for example – each of which requires some combination of DNA-conjugated antibodies, custom imaging reagents, and/or custom hardware, such as a TIRF illumination system, multiExR can be performed with only commercially available reagents, needing no end-user reagent preparation, and microscopy equipment that is widely available in ordinary biology labs and facilities. Indeed, we are not aware of any works amongst the reviewer’s claim “Competing nanoscale multiplexing methods have recently emerged in other labs’ preprints,” that require no end-user reagent preparation, and no hardware that is not widely available – simply put, it does not seem that a typical biology group would be able to use these techniques, without significant further investment.

Regarding whether the manuscript is unpolished and hastily put together: regarding the word “unpolished,” we would be happy to defer to the editor’s judgment on this matter. Regarding “hastily put together” – this simply is untrue, as the manuscript has been worked on for over 12 months to date.

Specific comments include:

- 1) The authors should present evidence that the 20 antibodies are validated and specific.

We note that our method is valid and useful, in general, even without association with a specific validated antibody – as long as the end user uses an antibody that they themselves have validated, they should be fine. The reason is because, as stated in the last paragraph, our current manuscript is not making a fundamental chemical innovation, but rather connects two existing protocols in optimized form – ExR, and serial immunohistochemistry. That said, we do care strongly that our current data be as valid, trustworthy, and accurate as possible. Accordingly, we provide the evidence the reviewer has requested, at a level that is standard in our field. The antibodies used in this experiment were purchased from commercial vendors. Our criteria for purchasing antibodies were, in order of priority: knockout or knockdown validation in tissue or cell lines, high user rating on vendor websites, multiple references for publications using the antibody, and vendor-provided images of Western blots and/or immunohistochemistry performed with the antibody. After purchasing the antibody, we also ensured that the resulting signal was present where expected, and absent otherwise. We note that these are standard practices in the field^{9,10}. We provide a summary of available validation data in the table below. The “list of antibodies” table (**Table 2**) has been updated in the text accordingly. Thus, we think that our antibodies have been validated to the standards of the field. That said, we reiterate that our technology is antibody-agnostic. That is, there are no *a priori* limitations on the antibodies that can be used successfully with multiExR, provided they are validated. The kind and degree of validation needed by an end user will ultimately be determined by the scientific question at hand – for example, some questions will require validation far beyond what is typically done (e.g., verifying post-translational modification specificity) and some will require less (e.g., if the exact isoform of a protein is not of interest). Here, we do a little bit of such scientific question-oriented validation – for example, we use multiple independent antibodies against beta amyloid, yielding similar results. This kind of validation, of course, would not be possible for a target with only one trusted antibody. We have added a note to the effect that validation should be customized to the scientific question at hand, in the Discussion section.

Table 2. List of antibodies

Primary / Secondary	Target	Host	Vendor	Product number	Dilution	Validation
Primary	Cav1.2	Guinea pig	Synaptic Systems	152 205	0.1805555 6	P ¹³ , VI
Primary	RIM1	Rabbit	Synaptic Systems	140 003	0.1805555 6	P ⁴⁴ , VI, KD

Primary	PSD95	Mouse	Thermo Fisher	MA1-046	0.1805555 6	P ²³⁰ , VI, KO
Primary	PSD95	Rabbit	Cell Signaling Technology	CST3450S	0.1805555 6	P ²⁹⁵ , VI, R
Primary	SynGAP	Rabbit	Thermo Fisher	PA1-046	0.1805555 6	P ³¹ , VI, KO
Primary	Homer1	Rabbit	Synaptic Systems	160 003	0.1805555 6	P ¹⁴¹ , VI, KO
Primary	Homer1	Chicken	Synaptic Systems	160 006	0.1805555 6	P ²² , VI, KO
Primary	Bassoon	Guinea pig	Synaptic Systems	141 004	0.1805555 6	No longer in stock*
Primary	Shank3	Guinea pig	Synaptic Systems	162 304	0.1805555 6	P ²² , VI, KO
Primary	Gephyrin	Mouse	Synaptic Systems	147 011	0.1805555 6	P ²⁰⁶ , VI, KO
Primary	GFAP	Chicken	Abcam	ab4674	0.1805555 6	P ⁵²² , VI
Primary	GluA1	Rabbit	Abcam	ab31232	0.1805555 6	P ¹⁴³ , VI, KO, R
Primary	CaMKII	Mouse	Abcam	ab22609	0.1805555 6	P ⁷⁸ , VI
Primary	Synapsin1	Rabbit	Abcam	ab8	0.1805555 6	P ⁵² , VI, R
Primary	NMDAR1	Mouse	ThermoFisher	32-050-0	0.1805555 6	P ⁴⁸ , VI
Primary	VGlut	Rabbit	Synaptic Systems	131 011	0.1805555 6	P ¹⁸⁷ , VI, KO
Primary	NR2B	Mouse	antibodiesinc	75-101	0.1805555 6	P ⁹⁴ , VI, KO

Primary	GluA4	Rabbit	Cell Signaling Technology	#8070S	0.18055556	P ¹⁵ , VI, R
Primary	GluA2	Mouse	antibodiesinc	75-002	0.18055556	P ¹⁸⁹ , VI, KO
Primary	PLP	Rabbit	Abcam	ab28486	0.18055556	P ⁷⁸ , VI
Primary	Stargazin	Mouse	ThermoFisher	PIMA527645	0.18055556	VI
Primary	Stargazin	Rabbit	Cell Signaling Technology	#8511	0.18055556	P ⁵ , VI, R
Primary	GluA3	Mouse	ThermoFisher	32-040-0	0.18055556	P ⁸ , VI, KO
Primary	GluA3	Rabbit	Abcam	ab40845	0.18055556	P ¹⁹ , VI
Primary	RIM-BP	Rabbit	Synaptic Systems	316 103	0.18055556	P ⁹ , VI, KO
Primary	A β 42 (6E10)	Mouse	BioLegend	SIG39320	0.18055556	P ³³¹ , VI
Primary	A β 42 (12F4)	Mouse	BioLegend	SIG39142	0.18055556	P ³¹ , VI
Primary	A β 42 (D54D2)	Rabbit	Cell Signaling Technology	CST8243S	0.18055556	P ¹¹⁶ , VI, R
Primary	SMI	Chicken	Abcam	ab4680	0.31944444	P ¹¹⁸ , VI, R
Primary	SMI	Chicken	BioLegend	822601	0.18055556	P ⁹ , VI
Primary	Kv7.2	Mouse	Santa Cruz	sc-271852	0.18055556	P ⁷ , VI, R
Primary	Nav1.6	Rabbit	Abcam	ab65166	0.18055556	P ⁹ , VI, R
Primary	ErbB4	Rabbit	Cell Signaling Technology	CST4795	0.18055556	P ⁸¹ , VI

Primary	Elfn1	Rabbit	Synaptic Systems	448 003	0.18055556	P ² , VI, KO
Secondary	Mouse	Goat	ThermoFisher	A28175 (Alexa Fluor 488 nm)	1:200	N/A
Secondary	Mouse	Goat	ThermoFisher	A11031 (Alexa Fluor 546 nm)	1:200	N/A
Secondary	Mouse	Donkey	Biotium	20124 (CF 633 nm)	1:200	N/A
Secondary	Mouse	Donkey	ThermoFisher	A10036 (Alexa Fluor 546 nm)	1:200	N/A
Secondary	Rabbit	Goat	ThermoFisher	A11034 (Alexa Fluor 488 nm)	1:200	N/A
Secondary	Rabbit	Goat	ThermoFisher	A11035 (Alexa Fluor 546 nm)	1:200	N/A
Secondary	Rabbit	Donkey	Biotium	20125 (CF 633 nm)	1:200	N/A
Secondary	Rabbit	Donkey	ThermoFisher	A10040 (Alexa Fluor 546 nm)	1:200	N/A
Secondary	Guinea pig	Donkey	Biotium	20171 (CF 633 nm)	1:200	N/A
Secondary	Chicken	Goat	ThermoFisher	A11039 (Alexa Fluor 488 nm)	1:200	N/A

Secondary	Chicken	Donkey	Biotium	20168 (CF 633 nm)	1:200	N/A
-----------	---------	--------	---------	-------------------	-------	-----

P: publications, number of references in superscript, VI: vendor image(s), KO: knock-out, KD: knock-down, R: high user rating (4 or more stars out of 5).

*Its replacement, Synaptic Systems 141 318 is supported by KO, P, and VI.

2) 'a reference channel with only nanoscale features is difficult to align at the macroscale'.

What does this mean? Why is it difficult?

Consider a reference channel consisting solely of staining a synaptic protein. It would appear, to the eye, to be a field full of small punctate objects. It would be very difficult for an experimentalist to find the same field of view imaged in a previous round, as synapses are small, numerous, and somewhat uniform in size and shape. Imagine trying to find a given star in a moonless night sky, without any large-scale structure (such as a constellation) to orient you. In contrast, macroscale features can easily be matched by eye. Once an experimentalist has achieved macroscale alignment by locating approximately the same field of view in a subsequent imaging round as in a previous round, using macroscale features identified by eye, then nanoscale features (such as synapses) can be used to help sculpt fine adjustments to co-register the images. Combining multiple single-protein stains, some of which are nanoscale (against synapses), and some of which are macroscale (against blood vessels), into a single reference channel allows for a reference to be achieved that has features appropriate for macroscale alignment as well as nanoscale alignment. One can try to register images using only a macroscale reference channel, but adding in fine features reduces error from an average of ~45 to ~32nm in the primary validation dataset and from ~87 to ~66nm in the secondary validation dataset, as we found by comparing the mean registration errors in staining and stripping rounds. The density of features, and the uniqueness of the feature detection space, facilitate matching at all scales by both human and algorithm.

3) What is 'adequate feature density'?

We quantified adequate feature density in our primary validation dataset, as described in the main text: "More specifically, we found that a reference channel feature density of ~1.3% (fraction of total image volume occupied by feature volume in the reference channel, which is the normalized sum of all channels in a given round in **Fig. 2**, but in other datasets is the three-target channel described above, 95% CI [0.01110,0.01402], **Extended Data Table 1**), with both large (e.g., largest feature ~50% of total reference feature volume like lectin, 95% CI [0.3707,0.6136], **Extended Data Table 1**) and small (e.g., smallest features 2.832⁻³% of total reference feature volume like synaptic proteins, 95% CI [2.368 x 10⁻⁵, 3.297 x 10⁻⁵], **Extended**

Data Table 1) features present (**Extended Data Fig. 2a**), is sufficient for accurate registration across rounds with 10-40nm round-to-round precision, in the most stringent form of the protocol.” We use the word “adequate” rather than “minimum” or “required,” because we do not show that lower feature densities would fail, or that higher feature densities would not be better.

4) Why is there not a neuronal marker to visualize the neuronal dendrites and axons that make these synaptic contacts?

Inclusion of a neuronal marker to visualize neuronal dendrites and axons alongside synaptic proteins is not necessary to substantiate the conclusions of our paper, which focuses on describing and validating a new technology. Indeed, we have based our stains on extensive prior work describing methods for super-resolution imaging of synapses, which also did not include such a stain^{1,11-13}. We note that including such a marker in multiExR experiments is feasible, and depending on the scientific question at hand, future users of multiExR could easily incorporate these and other stains.

5) Fig. 2 Round 1 and Round 2 show visible differences in the ratio between SynGAP and Bassoon staining. If this is an artifact of stripping, it needs to be addressed.

We agree with the reviewer that there is a visible difference in the ratio between SynGAP (red) and Bassoon (green) staining, between rounds 1 and 2 in **Fig. 2a**. After further examination, we observed that the Bassoon signal intensity, relative to SynGAP, increased markedly after the first round of stripping, and remained stable in subsequent rounds. The effect of the first round of stripping, effectively, is an antigen retrieval step – the harsh denaturation conditions (95C, in denaturation buffer, with beta-mercaptoethanol) used in antibody stripping helps reveal the epitope targeted in Bassoon, but not in SynGAP, for better downstream staining. To quantify this increase, we calculated the mean signal intensity (in background subtracted images) of pixels in synaptic puncta within manually-identified synaptic ROIs, and found this to increase in the second and third staining rounds (**Extended Data Fig. 2ci**), consistent with an antigen retrieval effect. In contrast, the absolute intensity of SynGAP staining, and of Bassoon staining on rounds beyond the third round, decreased somewhat steadily with successive rounds of stripping and re-staining, which suggests a general process of epitope staining efficacy decline occurs during harsh stripping conditions (**Extended Data Fig. 2cii**). However, the signal-to-noise ratio, number of puncta and puncta volumes (**Fig. 2d-e**, **Extended Data Fig. 2b**) were stable across rounds, demonstrating that while absolute intensity may vary between rounds of staining, detection of biologically meaningful objects was maintained, at least with the degree of antigen retrieval here demonstrated (one could imagine, if a signal were to go from

extremely dim to extremely bright, the number of puncta or puncta volume could change). In particular, it is striking that although absolute signal drops by a factor of 2 or 3, signal-to-noise (computed as signal of the object divided by signal of the background) stays constant – meaning changes in epitope staining efficacy may apply equally to background staining as to object staining. This is reassuring, and consistent with our past work (e.g., see **Fig. 4** of our paper¹⁴). The absolute intensity of an immunostained protein can be highly variable, of course, and can depend on many experimental factors, some of which are controllable and some of which are not. That said, even if staining intensity is of interest, in many cases, an experimentalist will be comparing an intensity across conditions, rather than evaluating the absolute intensity of a stain in one condition. For all these reasons, we are careful not to claim that absolute intensity is the same over multiple rounds of stripping and staining. We have added sentences to the main text of the manuscript to this effect.

6) Which brain region is shown in Fig.3? How is the brain sliced? Some basic information should be at least mentioned in the main text/figure caption to help understand the experiment.

It is the somatosensory cortex. We now have added this information to the figure caption. Information about slice preparation was previously provided in the Methods section, in the section “Brain tissue preparation.”

7) Why does the observation of AMPAR aggregation with amyloid require nanoscale imaging? The authors may want to present evidence that this cannot be seen by diffraction-limited confocal microscopy of non-expanded samples.

In previous work, we showed that beta-amyloid nanoclusters could not be seen with diffraction-limited confocal microscopy of non-expanded samples, or even with pre-expansion staining of samples that are then expanded¹. We have added a clarifying statement to this effect in the main text. Thus, visualization of such nanoclusters relies on the decrowding effect afforded by post-expansion staining. Finally, as shown in **Fig. 3a-b**, **Fig. 4 a** and **f**, and detailed in **Extended Data Table 8**, the beta-amyloid nanoclusters we see are nanoscale, measuring on average 0.0002563 (for 12F4) to 0.0002767 (6E10) cubic microns, the cube root of which is in the ~65nm range. Such structures would not be visible with diffraction-limited confocal microscopy, which has lateral resolution in the 180-250nm range¹⁵. Finally, we note that we had to screen through many targets to discover this AMPAR result - we screened over 20 different proteins for co-localization with amyloid. Thus, the decrowding effect of ExR was required to obtain this result, and the multiplexed nature of ExR was, if not invaluable, extremely helpful.

8) It is difficult to evaluate the (co)staining of different synaptic proteins in Fig.3&4&5. In particular, multiple proteins within the same field of view share the same pseudo-colors, which is confusing! The authors may want to show a panel of individual synapses, each with individual channels of synaptic proteins.

Multiple proteins within the same field of view were actually not sharing the same pseudo-color – we reused the same colors in each panel, for a different set of antibodies, but within any given panel, each protein had a different pseudo-color. In addition, what the reviewer is asking for, is actually already there – **Fig. 3a**, for example, already showed a small number of synapses, with each channel shown separately, and almost all of **Fig. 4a** was showing individual channels. In **Fig. 5a** and **a**, almost all of the panels were showing individual channels.

9) The vGluT1 signal in Fig.5(ii) does not seem to colocalize with any of the synaptic markers. One would also expect that the vGluT would fill up much of the presynaptic bouton around the Bassoon signal. Instead, vGluT looks like scattered puncta in the image.

Upon revisiting these images, we noticed the labels for the vGluT1 and GluA2 channels were swapped. Thus, the reviewer was commenting on the properties of GluA2 staining, which should not fill the synaptic bouton. We are grateful to the reviewer for pointing this out. We have corrected this mistake. Just in case we had made any other such errors - we checked all the other labels, throughout all the panels of the paper, and found that one other swap inadvertently occurred in Figs. 3 and 4 between GluA3 and NR2B. We have corrected this mistake. vGluT1 signal was relatively more diffuse compared to other proteins, after maximum intensity projection of a 3-dimensional image volume. However, examining a single z-plane revealed vGluT1 colocalization with other synaptic proteins, as expected. We have added an additional figure to illustrate this (**Extended Data Fig. 5**). We have also added text to reference this figure in the body of the manuscript.

Reviewer #2 (Remarks to the Author):

The authors reported a super resolution imaging method named multiplexed expansion revealing (multiExR), which enables high-fidelity visualization of >20 proteins in the same specimen, through serial rounds of staining and imaging. multiExR exhibits round-to-round registration error as low as 10-40 nm. The authors precisely mapped 23 proteins in the brain of 5xFAD Alzheimer's model mice, and found reductions in synaptic protein cluster volume, as well

as co-localization of specific AMPA receptor subunits with amyloid-beta nanoclusters. They also visualized 20 synaptic proteins in specimens of mouse primary somatosensory cortex. This work is a good example of how expansion-based microscopy could contribute to the pathological analysis in practical applications. The authors provided quite sufficient data to support their idea. I only have a few suggestions.

We thank the reviewer for the appreciative comments.

1. There are too many parentheses in the Results section, making it hard to read. Please rearrange those sentences.

We have rearranged several sentences to minimize the use of parentheses in the results section, except where appropriate to provide statistical information or figure or table references. We hope this increases clarity for the reader. We would be open to additional readability recommendations from the editor, if needed.

2. In the Main text section, adding some more introduction on expansion microscopy would be helpful for readers who are not that familiar with this field.

We thank the reviewer for this suggestion. We have added the following text to the introduction: “Expansion microscopy is a form of light microscopy that benefits from physical expansion of specimens, via chemical introduction of a densely permeating swellable hydrogel throughout a biological sample. Biomolecules or labels of interest are covalently anchored to the hydrogel. The specimen is then chemically softened, and then water is added, causing the hydrogel-specimen composite to swell in an even fashion (typically 4x, although more recent protocols support 10x, and iterating the procedure can yield 20x). The net effect is that the light microscope has an effectively increased resolution, even beating the diffraction limit of light microscopes². Early forms of expansion microscopy focused on labeling biomolecules before expansion³.”

3. Did the stripping process affect the reference channel? Please add more explanation on this issue.

To assess whether the stripping process affected the reference channel, we measured the mean signal intensity of the maximum intensity projection of the preprocessed, registered lectin channel from the primary validation dataset (**Fig. 2**) for each field of view and each staining and stripping round. The results are displayed in the plot below, now included in **Extended Data Fig. 2d**, with statistics summarized in **Extended Data Table 3**.

These results demonstrate that while the mean signal intensity in the single-channel lectin reference channel does appear to decline after the first round, the magnitude of the mean signal intensity is relatively stable in later rounds (e.g., rebounding a bit upwards, in round 3). (Note well - since we re-stained for lectin during the stripping rounds, to find the same field of view for imaging, the stripping rounds above can be thought of as additional staining rounds for this channel, except that we did not strip lectin again prior to the subsequent staining round.) While there was significant bouncing-around in mean signal intensity between rounds, there was no systematic pattern to the variation (**Extended Data Table 3**). As noted in response to Reviewer #1's Question #5, there are many reasons why random variation might occur in antibody staining, and thus absolute intensities are typically not the focus of immunostaining efforts. Rather, our focus in this paper is on the shape and volume of a stained region – metrics that are more robust, than intensity itself. Along these lines, even in the stripping rounds, when only the Lectin channel was re-stained, there remained sufficient signal to register the images within 80nm on average, and often within 50nm on average (**Extended Data Table 2**). Thus, regardless of the magnitude of signal intensity in the reference channel, its functional integrity was maintained after stripping, to an extent sufficient to make meaningful conclusions and to align images.

4. In the Discussion section, the authors said “map many proteins at once”, “map ~ 20 proteins at once”. This is inappropriate because multiplexing is realized in a sequential way.

We thank the reviewer for noting this. We have changed “at once” to “in the same field of view” to improve accuracy.

5. In the Discussion section, the comparison of multiExR with other super resolution imaging methods need more thorough discussion, especially for the comparison with Exchange-PAINT. The statement “but requires specialized reagents and equipment” is not convincing.

We have added the following text to the Discussion section to further substantiate our claim. “Another super-resolution method, Exchange-PAINT^{29,30}, a variation of DNA-PAINT³¹, can in theory be multiplexed to image more than four proteins in a single sample, but has not yet been demonstrated with 3D imaging in tissues. 3D DNA-PAINT imaging of organelles in cell cultures has been achieved^{32,33}. However, these studies were only shown with 2-3 channel multiplexing in cell cultures. Narayanasamy et al. demonstrated Exchange PAINT for super-resolution imaging of synapses in tissue³⁴. However, this approach was only shown with 2-dimensional images and is limited by the number of available secondary antibodies with compatible species. Another approach, multiplexed automated serial staining stochastic optical reconstruction (maS3STORM), demonstrated 3D super-resolution imaging of 16 targets in CNS tissue, but requires a dSTORM-capable microscope³⁵. Yet another method, molecule anchorable gel-enabled nanoscale imaging of fluorescence and stimulated Raman scattering microscopy (MAGNIFIERS) achieved 8-plex 3D nanoscale imaging in a mouse brain slice, but requires a Raman microscope³⁶. All of these approaches require custom DNA-conjugated antibodies, custom imaging reagents, and/or advanced hardware such as TIRF illumination systems, which are not readily available in most biology and neuroscience laboratories. By comparison, multiExR can generate 3-dimensional, 20+ channel, super-resolution images of proteins in tissue sections, using widely available reagents and an inverted confocal microscope.”

6. Line 460, please add more explanation on how to realize that “the same field of view is imaged again”.

The same field of view is located and re-imaged through the following process. In the first round of staining and imaging, we obtain a large (~500um x 500um, covering most of the gel) mosaic image of the reference channel with a 4x objective, followed by an image of a smaller region of interest with a 10x objective. These mosaic images of the reference channel are used to guide the experimentalist in finding the same field of view, at increasing magnification, in later rounds, using the same reference channel. We have added this information to the Methods section and a reference to the Methods in the figure caption.

7. Figure 3 and Figure 5, in the enlarged images, scale bars are missing.

The scale bar for these images was found in composite 1. Because the enlarged images were extracted from the larger fields of view, for which a scale bar was included, additional scale bars are not needed. Nevertheless, we have added extra scale bars to help orient the reader.

8. Figure 4, in the merged images, there seems to be obvious displacement between the blue channel and others. Please add some discussion on this issue.

The dark blue channel in the original submission, was the composite Lectin/SMI/GFAP reference channel – as a mixture of stains, it doesn't really have a biological meaning. One possibility is that the displaced component was the part of the reference channel reflecting neurofilament staining of axons with SMI, which appeared offset with regard to other stains. This is consistent with the known biology - we do not expect colocalization between axons and synaptic proteins or beta-amyloid nanoclusters. Given that the composite reference channel is not biologically meaningful, we removed it in the new manuscript, to minimize confusion.

The manuscript can be accepted if the above issues have been properly dealt with.

Reviewer #3 (Remarks to the Author):

Kang et al. introduced a method named multiplexed expansion revealing (multiExR), which achieved a resolution of approximately 20nm and the ability to detect roughly 20 proteins in a single sample. The team successfully mapped 20 synaptic proteins in a sample from the mouse primary somatosensory cortex. Notably, the study illustrated the co-localization of GluA2 and GluA4 glutamate receptor subunits with amyloid-beta nanoclusters in Alzheimer's model mice. This accessible and user-friendly method employs commonly available reagents and standard microscopes, indicating significant promise for future applications, especially with rare and valuable samples.

Two notable innovations in the study are worth highlighting: 1) the method's ability to sustain reliable signals even after up to 7 rounds of stripping and re-staining, and 2) the implementation of a 3D scale-intensity feature transform-based global registration algorithm enabling both global and local alignment of multiExR imaging. These two features enabled multiplexed super-resolution imaging of brain tissue using standard staining reagents and microscopes.

We thank the reviewer for the appreciative comments.

Consequently, the following concerns primarily revolve around these two points:

1. Insufficient stripping

This method made use of beta-mercaptoethanol-containing denaturation buffer to break the antibody disulfide bonds. It is unclear if all residual antibodies are effectively removed or if they might impact subsequent rounds of staining or epitope accessibility through accumulation.

We agree with the reviewer that inadequate stripping would impact subsequent rounds of staining. We have presented evidence to demonstrate effective stripping in **Fig. 2** and **Extended Data Fig. 2**. In summary, we show a qualitative and quantitative near absence of detectable synaptic protein after stripping, each round that we checked, in our validation experiment (**Fig. 2c**, **Extended Data Fig. 2b**; average of 821 Bassoon / 748 SynGAP objects in the staining round vs. 23 Bassoon / 42 SynGAP objects in the stripping round), which we support with an independent validation experiment (**Extended Data Fig. 3**). Furthermore, if antibodies from previous rounds were not sufficiently stripped, we would expect to see signal from those antibodies persist in later rounds, when other antibodies are being applied. Indeed, no such effect was seen. For example, in the 5xFAD vs. WT experiment (**Figs. 3-4**), PLP, a myelin protein, was stained in round 5 with a rabbit host antibody. Homer1 (round 6), D54D2 (round 7), GluA4 (round 8), PSD95 (round 9), SynGAP (round 10), and Stargazin (round 11) were all stained with rabbit host antibodies in subsequent rounds, but we did not see myelin staining in these channels.

1). Figure 2a raises questions about the persistence of Bassoon signal within lectin after round 1 stripping, in contrast to its absence after round 2 stripping. Would there be a possibility that there might be insufficient stripping for larger structures?

We are not sure why there is persistent Bassoon staining on the blood vessel after stripping round 1, but not after stripping round 2 (**Fig. 2a**). We speculate that insufficient stripping may be more likely to occur for “stickier” structures like blood vessels, where there may be more non-specific binding of Fc fragments, as we did not observe insufficient stripping outside of blood vessels. We observed non-specific staining in blood vessels, for some proteins, in all datasets (**Fig. 3a-b**, **Fig. 5a**, **Extended Data Fig. 3a**, and **Extended Data Fig. 6**). We note this in the main text.

2). In Figure 2c, the number of Bassoon detected in round 1 is lower than round 2 but this is not reflected in the representative images shown in Figure 2a and 2b. The discrepancy needs further clarification.

The number of Bassoon puncta detected in round 1 is lower than in round 2 for all fields of view (rounded average of 588 vs. 879 objects, respectively). Therefore, it is not possible for us to have chosen a field of view that is not representative of this difference. By eye, there is an increase in the relative magnitude of green puncta in the round 2 image (and later images) in **Fig. 2a-b**. To quantify this increase, we calculated the mean signal intensity (in background subtracted images) of pixels in synaptic puncta within manually-identified synaptic ROIs and found this to be increased in the second and third staining rounds and decreased afterwards (**Extended Data Fig. 2ci**), consistent with an antigen retrieval effect (as described in the answer to Reviewer #1, Question #5). However, the number of puncta and puncta volumes (**Fig. 2d-e**) are quite stable across rounds, demonstrating that while absolute intensity may vary between rounds of staining, detection of biologically meaningful objects is maintained, at least for the stains shown here. We have added a new extended data figure panel (**Extended Data Fig. 2c**) and new text to the manuscript to explain this.

2. Backgrounds caused by multiple rounds of stripping

1) When residual antibodies accumulate over rounds of staining and stripping, it would be expected to observe a slight decrease in signal as the antibodies might block access for the next round of staining. This can be observed in Figure 2a. However, in Extended Data Figure 2d, the signal in the reference channel appears to be even higher in rounds 5, 6, and 8. More notably, the shape of the structure in the reference channel is not very consistent, particularly after rounds 4 and 9 of stripping. This inconsistency is not in line with the registration error shown in Extended Data Figure 2c.

As noted in our previous response, and demonstrated by our analysis of synaptic ROIs in our primary validation dataset (**Fig. 2c-d, Extended Data Fig. 2d-e**), absolute intensity of an immunostained protein is highly variable and depends on many experimental factors, some of which are controllable and some of which are not. For these reasons (and see previous comment, and comments referenced therein), we were careful not to claim that absolute intensity is the same over multiple rounds of stripping and staining. Regarding inconsistency in the shape of the structure of the Lectin reference channel, the magnitude of the inconsistency was actually quite low, as shown in the below composite overlay of maximum intensity projections of the reference channel in the rounds mentioned by the reviewer. This is further evidenced, quantitatively, by the low registration errors shown in **Extended Data Fig. 3b** and summarized in **Extended Data Table 4**. Note well, these registration errors were low, even though (for the example dataset cited by the reviewer, and shown below), we actually varied the stain across rounds – removing the SMI, GFAP, and Homer stains with stripping, and re-staining with Lectin only to locate and image the same field of view in stripping rounds. For all other datasets, except where noted, the reference channel was kept constant across all rounds

(usually Lectin/GFAP/SMI; see **Extended Data Tables 5, 10, and 12**). One can try to register images using only a macroscale reference channel such as Lectin, but adding in fine features reduces error from an average of ~ 45 to ~ 32 nm in the primary validation dataset and from ~ 87 to ~ 66 nm in the secondary validation dataset, as we found by comparing the mean registration errors in staining and stripping rounds.

Composite maximum intensity projections of reference channels in secondary validation dataset field of view. Cyan: round 1, reference channel (Lectin/SMI/GFAP/Homer). Magenta: round 4, post-stripping (Lectin only). Yellow: round 9, post-stripping. Scale bar, 500nm (Lectin only).

2) In the analysis of signal-to-noise ratio, is the same threshold applied to every round of staining?

No, we allowed the absolute value of the threshold to vary, due to aforementioned variability in absolute signal intensity. The threshold was set at the 99.5th percentile intensity value for all images separately in all rounds of staining. This information, along with methods describing the calculation of the signal-to-noise ratio (SNR) presented in Fig. 2b were provided in the Methods section: “In order to determine whether nanoscale synaptic properties were maintained over seven rounds of stripping and staining, we quantified the number of puncta, mean puncta volume, and brightness (as measured by SNR) of manually-identified synaptic ROIs (**Fig. 2d**,

Extended Data Fig. 2d). 2-dimensional ROI boundaries were selected in Fiji using the rectangle tool. The ROI was cropped in 2 dimensions using the x- and y- boundaries from Fiji's ROI manager, with the z-boundary extending the entire 61-slice stack. Synaptic ROIs were processed as follows: conversion to grayscale, binarization using a threshold at the 99.5th percentile intensity value, and size filtration to remove puncta less than 30x30x30 nm³, which are likely noise. Synaptic protein puncta were defined as 3D connected components of the filtered, binary image volume (pixel connectivity of 26). Puncta volume was calculated from the binary mask volume using MATLAB's "regionprops3" function, multiplied by an average voxel size conversion factor of 1.2073×10^{-6} um³ per voxel (weighted average of x, y, and z spatial sampling rates). SNR was calculated as the mean intensity in the masked region (within synaptic protein puncta) divided by the mean intensity in the inverse of the masked region (within the background)." This has been updated in the revised manuscript to read: "In order to determine whether nanoscale synaptic properties were maintained over seven rounds of stripping and staining, we quantified the number of puncta, mean puncta volume, and brightness (as measured by absolute intensity and SNR) of manually-identified synaptic ROIs (**Fig. 2d-e, Extended Data Fig. 2b-c**). 2-dimensional ROI boundaries were selected in Fiji using the rectangle tool, based on the presence of both Bassoon and SynGAP staining in the first round. The ROI was cropped in 2 dimensions using the x- and y- boundaries from Fiji's ROI manager, with the z-boundary extending 15 slices in each direction from the center plus one frame. Synaptic ROIs were processed as follows: conversion to grayscale, binarization using a threshold at the 99.5th percentile intensity value, and size filtration to remove puncta less than 20x20x20 nm³, which are likely noise. Synaptic protein puncta were defined as 3D connected components of the filtered, binary image volume (pixel connectivity of 26). Puncta volume was calculated from the binary mask volume using MATLAB's "regionprops3" function, multiplied by an average voxel size conversion factor of 1.2073×10^{-6} um³ per voxel (weighted average of x, y, and z spatial sampling rates). SNR was calculated as the mean intensity in the masked region (within synaptic protein puncta) divided by the mean intensity in the inverse of the masked region (within the background). Mean absolute intensity was calculated as the mean intensity of pixels within the masked region."

3) Is there a recommended sequence for different proteins to be stained and stripped, like most abundant or less abundant protein first stained?

We do not provide a blanket recommended sequence for the staining of different targets. The prioritizing of protein targets in rounds may depend on the purpose of the study, the expression level of the protein targets, and the quality and host species of the antibodies. It may be helpful to image one or more structural/marker proteins in earlier rounds to choose the most appropriate fields of view for imaging in later rounds. Additionally, low-expressing protein

targets and/or antibodies that yield low signal to noise might be better suited for earlier rounds, where the chance of signal degradation is lowest (although note the observation above, that some epitopes may benefit from an antigen retrieval effect of stripping), while high-expressing protein targets and/or antibodies that yield high signal to noise might be better suited for later rounds. We have added similar text to the caption of **Extended Data Fig. 1**.

3. Registration.

1) It is not clear to me why sometimes they can achieve 30nm mean registration error but later only 70nm or 100nm. Can the author list out the exact conditions when different level of registration accuracy can be achieved. As 50-100nm mean registration error is still quite significant considering the desired resolution of ExR is 20nm. Is this caused by nonlinear distortion due to stripping and re-staining, the quality of the reference channel after multiple rounds of restaining, or other computational limitation?

The conditions that led to each registration accuracy level were discussed in the main text associated with each figure in our first submission. In summary, we provided information on the reference channel and registration algorithm used for each dataset, with reference to **Extended Data Fig. 1**. Exact registration errors for each round pair for each field of view in each dataset are provided in **Extended Data Tables 2, 4, 6, 11, and 13**. In our first submission, we provided an explanation for differences in registration error between datasets, including between the two validation datasets, in the main text: “We speculate that the higher registration error in this secondary validation dataset was due to relative signal intensity differences between the proteins co-stained in the single reference channel (**Extended Data Fig. 2c**), which reduced the quality of feature detection across scales during SIFT-based registration.” Finally, to support multiExR users in choosing a reference channel and registration method that supports their needs, we created schematized guidelines in **Extended Data Figure 1**. Though we cannot guarantee an exact registration error value for each set of conditions, we provide an estimate based on our experience in **Extended Data Fig. 1B**.

Regarding the reviewer’s second question, we provided a description of the factors leading to the observed registration error in the first sub-heading of the results section: “Computational registration with nanoscale precision of multiExR images taken across multiple rounds of staining and imaging initially posed great difficulty, despite innovations such as our multiscale reference stain, and the aforementioned improvements in sample stabilization during expansion and imaging. multiExR gels are free-floating, because immobilization of gels used in standard ExM imaging reduces antibody stripping efficiency, and gels would often detach anyway from glass surfaces during strong antibody stripping treatment. However, free-floating gels exhibit more degrees of freedom and variability between rounds of imaging than

immobilized gels, even with careful experimental handling. Indeed, translation, rotation, scaling, and shearing, can be heterogeneous even within a given gel. Furthermore, due to the highly expanded nature of ExR gels, and the dilution of tissue structure due to such expansion, features can be sparser than ideal for registration. Finally, slight variation in signal-to-noise ratio (SNR), perhaps due to the stochasticity of antibody binding at the nanometer scale, means that even identical staining and imaging conditions across rounds can lead to slightly, but significantly, different images in the reference channel –perhaps a fundamental issue for any imaging protocol involving antibody staining.”

2) What are the criteria of the protein to be chosen as reference channel?

Criteria for a protein (or set of proteins) to be chosen as a reference include high signal-to-noise ratio and adequate feature density (see reply to Reviewer #1, comments 2 and 3, for details). It may also be useful to choose a reference protein with some structural, morphological, or cell type information that guides the viewer to appropriate subvolumes for detailed imaging. We have added this text to the main text.

Minor Concerns:

1. It is essential to acknowledge that the quality of imaging is heavily reliant on antibody performance. In the interest of user convenience and the future application of this method in the broader field, can the authors please list the antibodies and corresponding conditions that were attempted but did not yield expected results?

We strongly agree, and we thank the reviewer for this suggestion. “Finally, it is important to note that multiExR image quality depends on antibody performance. To assist future users, we have provided a list of antibodies that yielded negligible signal with multiExR in the region imaged (**Extended Data Table 14**)”. We have added the following text to the discussion, and the corresponding information (below) in **Extended Data Table 14**.

Primary / Secondary	Target	Host	Vendor	Product number	Dilution Factor
Primary	mGluR5	Chicken	Aveslabs	ER5	1:200

Primary	Adam22	Mouse	Antibodies inc	75-093	1:200
Primary	GABA-B	Guinea pig	Millipore Sigma	AB2256	1:200
Primary	CACNA1G	Rabbit	Fisher Scientific	50-173-1816	1:200
Primary	CACNG8	Rabbit	Alamone labs	ACC-125	1:200

2. Extended Data Figure 1c lacks clarity in terms of what (A), (B), and (C) refer to.

We have fixed this mistake in **Extended Data Fig. 1c**.

3. The arrangement of Extended Data Figure 2 is confusing, as 2a and 2b share the same image dataset with Figure 2, whereas Extended Data Figure 2c and 2d belong to a separate image dataset, making it difficult for readers to follow.

We have separated the figure into **Extended Data Figure 2** and **Extended Data Figure 3** to avoid this confusion.

4. Figure 3a, 3b, and 5a suffer from low image resolution, making it challenging to discern between signal and noise.

Given the restrictions on figure sizes in an 8.5x11" document form, we are unable to expand the figure further. We think the size of the fields of view we have shown in **Figs. 3a, 3b, and 5a** are optimal for illustrating the size of the field of view captured at 40x magnification with ~18x expansion afforded by multiExR, while the zoomed insets are optimal for viewing smaller features. Images in the final publication will be higher resolution, and the online version will allow readers to zoom into figure panels, which will ameliorate this issue. Additionally, the registered image volumes and corresponding metadata are available for download from the Harvard Dataverse, in full, as mentioned in the manuscript. Any reader can download the images and render them as they see most instructive. We are open to editor suggestions regarding what the journal might offer in the way of raw data sharing.

References

1. Sarkar, D. *et al.* Revealing nanostructures in brain tissue via protein decrowding by iterative expansion microscopy. *Nat. Biomed. Eng.* 2022 69 **6**, 1057–1073 (2022).
2. M'Saad, O. *et al.* All-optical visualization of specific molecules in the ultrastructural context of brain tissue. *bioRxiv* 2022.04.04.486901 (2022) doi:10.1101/2022.04.04.486901.
3. Louvel, V. *et al.* iU-ExM: nanoscopy of organelles and tissues with iterative ultrastructure expansion microscopy. *Nat. Commun.* 2023 141 **14**, 1–18 (2023).
4. Ku, T. *et al.* Multiplexed and scalable super-resolution imaging of three-dimensional protein localization in size-adjustable tissues. *Nat. Biotechnol.* 2016 349 **34**, 973–981 (2016).
5. Jungmann, R. *et al.* Multiplexed 3D cellular super-resolution imaging with DNA-PAINT and Exchange-PAINT. *Nat. Methods* 2014 113 **11**, 313–318 (2014).
6. Wang, Y. *et al.* Rapid Sequential in Situ Multiplexing with DNA Exchange Imaging in Neuronal Cells and Tissues. *Nano Lett.* **17**, 6131–6139 (2017).
7. Klevanski, M. *et al.* Automated highly multiplexed super-resolution imaging of protein nano-architecture in cells and tissues. *Nat. Commun.* **11**, (2020).
8. Shi, L. *et al.* Super-Resolution Vibrational Imaging Using Expansion Stimulated Raman Scattering Microscopy. *Adv. Sci.* **9**, (2022).
9. Hewitt, S. M., Baskin, D. G., Frevert, C. W., Stahl, W. L. & Rosa-Molinar, E. Controls for Immunohistochemistry. <http://dx.doi.org/10.1369/0022155414545224> **62**, 693–697 (2014).
10. Bordeaux, J. *et al.* Antibody validation. *Biotechniques* **48**, 197–209 (2010).
11. Chen, F., Tillberg, P. W. & Boyden, E. S. Expansion microscopy. *Science (80-.).* **347**, 543–548 (2015).
12. Dani, A., Huang, B., Bergan, J., Dulac, C. & Zhuang, X. Superresolution imaging of chemical synapses in the brain. *Neuron* **68**, 843–856 (2010).
13. Tang, A. H. *et al.* A trans-synaptic nanocolumn aligns neurotransmitter release to receptors. *Nat.* 2016 5367615 **536**, 210–214 (2016).
14. Valdes, P. A. *et al.* Improved immunostaining of nanostructures and cells in human brain specimens through expansion-mediated protein decrowding. *Sci. Transl. Med.* **16**, eabo0049 (2024).
15. Schermelleh, L., Heintzmann, R. & Leonhardt, H. A guide to super-resolution fluorescence microscopy. *J. Cell Biol.* **190**, 165–175 (2010).

Reviewers' Comments:

Reviewer #1:

Remarks to the Author:

The authors have fixed some mistakes and typos during revision. I engaged a senior PhD student with extensive ExM experience for this review round. For them, the revised manuscript contains a lot of work but, unfortunately, is loaded with dreary technical details, obtuse writing, and not suitable for a broad readership.

Main comments:

1. We have serious doubts about how practical the method is especially when one wants to achieve super-resolution with low registration error. To reach the maximum potential presented in the paper, this method is limited by time-versus-resolution tradeoff. For example, the processing time for 11+ targets, as indicated by Extended Data Fig. 6, can take weeks when quality is prioritized. On the other hand, speed-prioritized protocols quickly reach registration errors ~100 nm, larger than the resolution promised by ExM.
2. Antibody stripping after ExM, as a major innovation of this work, is not sufficiently novel. One would not try to publish antibody stripping with dSTORM, PAINT, STED, etc, in Nat. Comm.

Minor comments:

- 1- Authors may want to elaborate on the biological context and relevance of their data (disease model, synaptic protein interactions etc), especially in Discussion.
- 2- The reasoning behind the choice of the reference channel markers is not clear. The authors should mention what each of these markers labels (it's hard to see or understand it in any of the images). Why did the authors not use the NHS-ester dye that is commonly used in expansion microscopy experiments for this channel?
- 3- Mistakes remain in figures:
 - a. Fig. 1 b) does not have the description for (iv), and there is a typo with the numbering of panels ii and iii.
 - b. In Fig. 2 f) the table is too small and unreadable. Perhaps the data can be included in the figure caption/text.
 - c. Fig. 3 c) Why does the registration error show a systematic increase only in the 5xFAD graph?
 - d. Fig. 3 We are confused about why the authors have chosen to analyze only the ROIs that don't overlap with beta-amyloid nanoclusters.
 - e. Fig. 5 in lines 650-651 the red arrow and blue arrow are the opposite of what is explained in the main text in lines 357-359.

4- Non-uniform use of the abbreviations and their full-length names in the manuscript.

From the authors' response to reviewer 2 comment 3, there seemed to be a significant loss of lectin staining within just 2 rounds of stripping. The authors argued that this is not a major concern because their analysis focuses on shape and volume of stained regions. However, loss of weak signal is a major concern esp. for super-res microscopy that promises single-protein resolution. This method favors regions with dense signals (e.g. protein clusters) over dispersed protein targets in the cytosol, which is an important caveat of MultiExR.

The authors' response to reviewer 2 comment 5 is not convincing because multiExR requires complex image registration pipelines, as indicated in the manuscript. It does not appear to be any simpler than existing methods. Furthermore, it seems to lead to inexplicable observations such as the one removed by the authors during revision in reviewer 2 comment 8.

Reviewer #3:

Remarks to the Author:

The authors have addressed most of our concerns.

Even though the authors discussed the variability of registration accuracy in detail, it is clear that the registration accuracy limited the resolution of MultiExR significantly. Considering the mean registration error is about 50-100nm, we would suggest the authors state the resolution of MultiExR more conservatively and honestly, rather than present it as if MultiExR shares the same resolution of ExR, such as in line 417 (MultiExR enables ~20 nm resolution imaging, with potentially very high multiplexing capacity, requiring only ordinary microscopes and common laboratory reagents).

Meanwhile, we still have concerns regarding the calculation of the signal-to-noise ratio. The rationale behind defining puncta smaller than $30 \times 30 \times 30 \text{ nm}^3$ as noise is unclear, which raises doubts about the reliability of the threshold setting. In Figure 4fi, GluA4 appears to fall within this size range but is still included in the quantification, highlighting inconsistencies in the threshold criteria.

We thank the reviewers for their constructive comments. We have performed further work to address all the remaining concerns of the reviewers. We hope the paper is now acceptable at Nature Communications.

The authors have fixed some mistakes and typos during revision. I engaged a senior PhD student with extensive ExM experience for this review round. For them, the revised manuscript contains a lot of work but, unfortunately, is loaded with dreary technical details, obtuse writing, and not suitable for a broad readership.

We have worked to improve the readability of the main text by breaking large chunks of text into smaller, more digestible paragraphs, so that complex concepts can stand alone. We have also worked hard to break up complex sentences, into simpler ones. Finally, we have also removed some in-text statistics, keeping only the reference to the corresponding Extended Data table, which contains the full statistics for the paper. However, given that this paper describes a new method, and how to use it, some technical details are unavoidable; we have tried to frame, for the reader, when they should pay attention to detail, and when it is optional. Hopefully this helps make the paper suitable for a broad readership, while still preserving the important technical detail needed by some audiences.

Main comments:

1. We have serious doubts about how practical the method is especially when one wants to achieve super-resolution with low registration error. To reach the maximum potential presented in the paper, this method is limited by time-versus-resolution tradeoff. For example, the processing time for 11+ targets, as indicated by Extended Data Fig. 6, can take weeks when quality is prioritized. On the other hand, speed-prioritized protocols quickly reach registration errors ~100 nm, larger than the resolution promised by ExM.

Regarding the reviewer's comment "We have serious doubts about how practical the method is especially when one wants to achieve super-resolution with low registration error": we agree that the method may not be practical for certain research applications, such as for large-sample size, confirmatory experiments, especially when 4 or fewer protein targets need to be imaged. For such experiments, classical ExM or ExR protocols may be more appropriate. As mentioned in the

discussion and demonstrated in **Figs. 3-5**, multiExR may be most useful as an early-in-a-project, hypothesis-generating, exploratory technique, to be performed a small number of times at the beginning of a study, to help direct the energies of investigators later in their study. This would be consistent with how other high-resolution multiplexed techniques are used (e.g. refs¹⁻⁴). We have added a note in the main text to this effect.

We agree with the reviewer that “To reach the maximum potential presented in the paper, this method is limited by time-versus-resolution tradeoff. For example, the processing time for 11+ targets, as indicated by Extended Data Fig. 6, can take weeks when quality is prioritized.” Indeed, we detail (though did not run a systematic set of experiments to test) this tradeoff in the form of a decision tree in **Extended Data Fig. 1**, to which we believe the reviewer is referring. Acquiring a large number of targets with low registration error is indeed a slow process, but a given scientific study might need this only once, at the beginning of a study, to suggest hypotheses to be tested through later causal perturbation or function-oriented studies. The timescale we report is similar to that of other cyclic immunostaining techniques⁵⁻⁷, due to fundamental limitations such as the need for multiple rounds of staining, imaging, and washing. Speed bottlenecks include overnight incubations for primary and secondary antibodies and time spent imaging. If speed is a priority, users could explore methods of shortening antibody incubation time (e.g., 2 hours at room temperature for secondary antibodies, though a shorter incubation time would need to be tested empirically for each primary and secondary antibody) or imaging a small, targeted, set of regions of interest to shorten imaging time. But given the nature of staining, imaging, and washing, such speed considerations would apply to any high-resolution cyclic staining method. We have added text to this effect in the Discussion section.

Regarding the reviewer’s comment “On the other hand, speed-prioritized protocols quickly reach registration errors ~100 nm, larger than the resolution promised by ExM”: we agree; this is why we did not follow the speed-optimized protocol for the datasets presented in the paper. We now recommend that the users should not follow the speed-optimized protocol, unless they really know that their reason for performing ~20x expansion can tolerate ~100nm registration error (e.g., for assessing the presence of many proteins at once, in the same sample, and their nanoscale relationship to cellular reference landmarks imaged on every round, but without regard for their relative organization to one another). Additionally, although 100nm is larger than the resolution promised by ExM, and therefore, one might consider using a multiplexed version of ExM with ~4x expansion (e.g., MAP⁸) or

dExPath⁹, it is unclear whether these methods guarantee a registration error of ~100nm or better, as they did not attempt to measure registration error. Indeed, the registration error for smaller expansion factor multiplexed datasets could be higher than for larger expansion factor multiplexed datasets, for any given offset in pixels (e.g., a 4-pixel error corresponds to ~35-40nm at 18x expansion in our dataset, but would correspond to ~160-165nm at 4x expansion).

2. Antibody stripping after ExM, as a major innovation of this work, is not sufficiently novel. One would not try to publish antibody stripping with dSTORM, PAINT, STED, etc, in Nat. Comm.

Regarding “Antibody stripping after ExM, as a major innovation of this work, is not sufficiently novel:” we agree with the reviewer that the basic idea to strip and restain antibodies is not new. However, as described in our paper, optimizing antibody stripping to be compatible with ExR was not trivial, and involved innovations including incorporating beta-mercaptoethanol into the stripping buffer to break the disulfide bonds of applied antibodies, using brief stripping times at high temperature (95 C for 1 hr) to maximize epitope preservation, creating a multiscale registration strategy to identify the same fields of view in sequential imaging rounds, and minimizing registration error through specific wet-lab and stain-design choices. Finally, as described in the text, optimizing and validating the software to register and analyze multiExR images was not trivial, and represents an innovation in itself – please note, despite many prior claims by expansion and other super-resolution microscopy papers of multiplexed antibody staining in a high-resolution context, we found only one paper that validated the nanoscale precision of their multiround registration³, a paper involving dSTORM imaging. Finally, the registered multiExR datasets shown reveal biologically novel findings, provoking new hypotheses, through their decrowding and multiplexing capabilities, and showing the value of the presented protocol.

Regarding “One would not try to publish antibody stripping with dSTORM, PAINT, STED, etc, in Nat. Comm.”, we note in this paper that multiExR could be implemented by any life science laboratory, without new equipment, meaning that the result is practical and useful in everyday biology. Other super-resolution methods require equipment that most biology labs do not have. We note that ExM has already been used in over 600 experimental papers and preprints – it is spreading very rapidly in biology. We do not know of any other super-resolution method that is spreading as quickly in biology, than ExM. Our hope is to extend that practicality, already widely appreciated in biology, to the domain of multiplexed high resolution protein mapping. We share all protocols freely, which should help with deployment of

such tools into the community.

Minor comments:

1- Authors may want to elaborate on the biological context and relevance of their data (disease model, synaptic protein interactions etc), especially in Discussion.

To the Discussion, we have added text to the last two paragraphs, to provide biological context for our results.

2- The reasoning behind the choice of the reference channel markers is not clear. The authors should mention what each of these markers labels (it's hard to see or understand it in any of the images). Why did the authors not use the NHS-ester dye that is commonly used in expansion microscopy experiments for this channel?

We chose Lectin, a marker of blood vessels, as the millimeter-to-micron-scale reference channel, because blood vessels are present throughout the brain parenchyma, and exhibit unique morphologies that allow a researcher to visually locate the same field of view for imaging across rounds. However, we found blood vessels alone did not provide sufficient nanoscale feature density for fine-scale registration. Thus, we added neurofilament and/or glial-process markers, SMI and GFAP respectively, as well as a synaptic scaffolding protein (Homer) to the same reference channel, to facilitate nanoscale feature identification and mapping across rounds. Each of these markers is expected to be abundant in the brain areas we imaged.

Users are not limited to our choice of reference channel. Indeed, if multiExR is applied outside of the brain, in other tissues, users will have to use a different reference channel, which will need to be validated and optimized. We think any abundant, bright (high signal-to-noise), and heterogeneous (i.e., unique features at multiple length scales, from nano to macro) stain could work as a reference channel. We have added the above two paragraphs to the Discussion section to aid the reader.

Regarding the reviewer's question, "Why did the authors not use the NHS-ester dye that is commonly used in expansion microscopy experiments for this channel?" – in principle, this could make for an excellent choice, and we note this in the Discussion section now. In practice, NHS-ester staining (e.g., a 1.5 hr incubation with 20 µg/mL NHS ester dye in 100 mM sodium bicarbonate solution (Sigma-Aldrich, catalog no.

SLBX3650) on a shaker at RT, and washing three to five times in PBST for 20 min each on the shaker at RT) was different enough from our antibody staining condition, that we were concerned that users would need to perform additional steps after antibody staining – potentially, on each round - adding time to an already-long protocol. Given our desire to keep protocol time to the minimum needed for doing good science, we found that our antibody-based reference channel strategy could achieve a small enough registration error, so we did not pursue NHS-ester staining for the reference channel.

3- Mistakes remain in figures:

a. Fig. 1 b) does not have the description for (iv), and there is a typo with the numbering of panels ii and iii.

We have made these corrections.

b. In Fig. 2 f) the table is too small and unreadable. Perhaps the data can be included in the figure caption/text.

We have made this change.

c. Fig. 3 c) Why does the registration error show a systematic increase only in the 5xFAD graph?

The mean registration error also increased steadily over rounds in most WT fields of view, as detailed in **Extended Data Table 6**, but not in the field of view shown in **Fig. 3**. We note that the mean registration error also increased over rounds for most 5xFAD fields of view. As noted in the figure caption, the registration error shown for the WT field of view in **Fig. 3c** corresponds to the WT field of view shown in **Fig. 3a**. We chose the field of view in **Fig. 3a** based on its low but still representative registration error (see below; 3rd lowest registration error of all WT fields of view) and high signal quality upon manual inspection. Importantly, the difference in registration error patterns noted by the reviewer does not affect the analysis shown in **Fig. 3** because the quantification of overall synaptic abundance in the whole field of view is not impacted by registration quality. Furthermore, registration error in the ranges achieved does not impact the analysis shown in the majority of **Fig. 4** because protein abundance (**Fig. 4b, d, e**) is calculated in manually identified beta-amyloid nanocluster ROIs, which are at least several hundred nanometers in linear

dimension, and centered around the protein of interest with space on either side. Therefore, offsets in the <50nm range will not shift the protein of interest outside of the ROI boundaries and impact quantification. For the volumetric overlap quantified in **Fig. 4c**, high-offset ROIs were manually excluded as described in order to reduce the impact of registration quality on these results.

Field of view	mean registration error (um)		
S3ROI1	0.039512		
S3ROI2	0.020314	shown in Fig. 3c	
S3ROI3	0.021478		
S3ROI4	0.017586		
S4ROI1	0.052906		
S4ROI2	0.043473		
S4ROI3	0.054354		
S4ROI4	0.015505	WT median	0.030495

d. Fig. 3 We are confused about why the authors have chosen to analyze only the ROIs that don't overlap with beta-amyloid nanoclusters.

For this analysis, we wanted to analyze putative synapses that did not contain amyloid, so that we could examine synapses in the Alzheimer's context that were not directly affected by the physical presence of amyloid – potentially they could be in a different state than when beta-amyloid was physically present in conjunction with synaptic proteins, which we describe at length in **Fig. 4**. One might speculate whether such synapses are “healthier,” for example - it would be of biochemical interest to see how synaptic proteins might differ in their state when they are near amyloid vs. far from amyloid. We have edited the main text to note our reasoning for this choice.

e. Fig. 5 in lines 650-651 the red arrow and blue arrow are the opposite of what is explained in the main text in lines 357-359.

We have fixed the caption of Fig. 5.

4- Non-uniform use of the abbreviations and their full-length names in the manuscript.

We cleaned up such usage, aiming to use abbreviations when they are commonly

used in the field, for all incidences after the first usage. For example, “beta-amyloid” was replaced by A β whenever possible.

From the authors’ response to reviewer 2 comment 3, there seemed to be a significant loss of lectin staining within just 2 rounds of stripping. The authors argued that this is not a major concern because their analysis focuses on shape and volume of stained regions. However, loss of weak signal is a major concern esp. for super-res microscopy that promises single-protein resolution. This method favors regions with dense signals (e.g. protein clusters) over dispersed protein targets in the cytosol, which is an important caveat of MultiExR.

The reviewer is correct in noting a loss of Lectin signal intensity after the first round of stripping (**Extended Data Fig. 2d**). The effect, while significant, is only appreciable during the first round, and does not decline much further on subsequent rounds, meaning that the effect does not get progressively worse, and thus may not affect most experiments, especially when the initial signal is very prominent (as it is, for lectin), and thus can tolerate a few-fold change in brightness after the first round. Indeed, ExR will most likely be used with fairly bright and bold signals: we are puzzled as to why the reviewer thinks that multiExR “promises single-protein resolution,” as we never make that claim. Indeed, we are very up front that our resolution is ~20 nm, much larger than a single protein. Furthermore, we are very up front that ExR is most useful for densely packed proteins, that would benefit from being separated from each other, to facilitate immunostaining – exactly the “dense signals (e.g., protein clusters)” that the reviewer says we favor. For a “weak signal” comprising “dispersed protein targets,” there might be less need for decrowding, all other things held equal, and thus other methods of expansion may suffice. We have clarified this in the main text.

The authors’ response to reviewer 2 comment 5 is not convincing because multiExR requires complex image registration pipelines, as indicated in the manuscript. It does not appear to be any simpler than existing methods. Furthermore, it seems to lead to inexplicable observations such as the one removed by the authors during revision in reviewer 2 comment 8.

Regarding the comment, “The authors’ response to reviewer 2 comment 5 is not convincing because multiExR requires complex image registration pipelines, as indicated in the manuscript”, we would like to distinguish between software complexity and hardware complexity. We argue that most users would prefer to run complex code (and indeed, some users may not consider running this code to be very complex at all) than to purchase (potentially at very great cost), set up, and/or

learn how to use a new microscope system. Furthermore, given the advent of new generative AI tools, understanding, modifying, and running code will only become easier for experimental biologists over time. Objectively speaking, multiExR has considerably reduced hardware complexity compared to other multiplexed nanoimaging methods, as detailed in the discussion section.

Regarding the reviewer's comment "Furthermore, it seems to lead to inexplicable observations such as the one removed by the authors during revision in reviewer 2 comment 8" – there seems to be a misinterpretation or miscommunication going on here. As described in our original response to reviewer 2, comment 8 (pasted below), the displacement between Lectin/SMI/GFAP, being a composite of many channels mixed together, and other channels, was not at all inexplicable – it was expected. Rather, this channel was removed because it was not relevant to the figure's meaning, and thus could have been potentially confusing to the reader. However, should any reader or reviewer want to examine these channels, they are available for download from the Harvard Dataverse at <https://dataverse.harvard.edu/privateurl.xhtml?token=3cd5c49f-12bc-4849-a016-8b925691e8c2>, and we would fully defer to the editor, if it would be preferred to put this channel back into the figure.

Quoting from our previous rebuttal:

Reviewer 2. 8. Figure 4, in the merged images, there seems to be obvious displacement between the blue channel and others. Please add some discussion on this issue.

The dark blue channel in the original submission, was the composite Lectin/SMI/GFAP reference channel – as a mixture of stains, it doesn't really have a biological meaning. One possibility is that the displaced component was the part of the reference channel reflecting neurofilament staining of axons with SMI, which appeared offset with regard to other stains. This is consistent with the known biology - we do not expect colocalization between axons and synaptic proteins or beta-amyloid nanoclusters. Given that the composite reference channel is not biologically meaningful, we removed it in the new manuscript, to minimize confusion."

Reviewer #3 (Remarks to the Author):

The authors have addressed most of our concerns.

Even though the authors discussed the variability of registration accuracy in detail, it is clear that the registration accuracy limited the resolution of MultiExR significantly. Considering the mean registration error is about 50-100nm, we would suggest the authors state the resolution of MultiExR more conservatively and honestly, rather than present it as if MultiExR shares the same resolution of ExR, such as in line 417 (MultiExR enables ~20 nm resolution imaging, with potentially very high multiplexing capacity, requiring only ordinary microscopes and common laboratory reagents).

As described in our response to Reviewer 1, comment 1, we encountered a time vs. resolution (via registration accuracy) tradeoff with multiExR, which is detailed in **Extended Data Fig. 1**. If users desire the ~20nm resolution of ExR and thus require 20nm or less registration error, they can choose a registration channel strategy that enables this, though it will take more time, or enable imaging of fewer protein targets. As stated in our response to Reviewer 1, above, we now recommend that users use the high-precision form of multiExR, unless they explicitly know they want to go with less precision. We agree with the reviewer that softening our claim about the resolution of multiExR would be appropriate, and have done so by revising the corresponding sentences to read: “20nm or larger , depending on the registration error achieved” or similar.

Meanwhile, we still have concerns regarding the calculation of the signal-to-noise ratio. The rationale behind defining puncta smaller than 30x30x30 nm³ as noise is unclear, which raises doubts about the reliability of the threshold setting. In Figure 4f(i), GluA4 appears to fall within this size range but is still included in the quantification, highlighting inconsistencies in the threshold criteria.

We agree with the reviewer that the GluA4 visible in **Fig. 4f(i)** may fall within the size range that is excluded in other analyses in the paper. The size filters used for each analysis varies, depending on the biological goal at hand. For clarification, we summarize our use of size filters in the revised manuscript below (all info is available in the Methods section):

- **Fig. 2c:** Whole field of view automated synapse counting: “size filtration with a lower limit of 50x50x50 nm³”
 - Note, this translates to 111 voxels
- **Fig. 2d-e:** within manually-identified synaptic ROIs: “and size filtration to remove puncta less than 20x20x20 nm³, which are likely noise”
 - Note, this translates to 8 voxels

- **Fig. 3e:** synaptic proteins between WT and 5xFAD mice (*not* manually-identified ROIs): “size filtration to include only puncta with volume greater than 100 voxels and less than 5000 voxels”
- **Fig. 4b:** “For beta-amyloid species, size filtration with a minimum volume of 50 voxels was applied.”
- We note that no size filter was applied for the analyses shown in **Fig. 4c-f** for GluA2 and GluA4. However, due to a typo in the code, we believe a filter size of 50 voxels was inadvertently applied for GluA1 and GluA3. We have fixed this mistake. To be consistent with the analysis in **Fig. 4b**, we have applied a size filter of 20 voxels for the analysis of GluA1-4 in **Fig. 4c-f**.

How were these voxel thresholds selected? Our selection of size filter for each of these analyses was based on empirical findings related to what produced a reasonable mask to exclude high spatial frequency noise based on visual inspection, as well as the expected size of a synapse (e.g., the synaptic cleft is about ~20 nm). Too strict a size filter (e.g., requiring puncta to be >150 voxels or 50-60 nm) leads to exclusion of puncta within actual synapses, while too permissive a size filter (e.g., <1 voxel or <10 nm) may be ineffective at removing high spatial frequency noise. We acknowledge that thresholding followed by filtration-based methods of puncta/synapse identification are, due to inevitable variability, bound to produce some false positives and false negatives. This is why we used manually-identified ROIs for detailed analyses within putative synapses or beta-amyloid nanoclusters, and decreased the size filter used on manually-selected ROIs to reduce the chance of excluding actual puncta in such regions, which we are confident contain the biological structures of interest.

To investigate the effect of our choice of size filter on our results, we conducted three parameter scans as detailed below.

- 1) We re-ran the calculation of signal-to-noise ratio in the validation dataset (**Extended Data Fig. 2b**), the measure that the reviewer was initially concerned about, with various size filters in the staining rounds. The size filters tested were chosen to span the range from 0 (no size filter) to double our *original* (first submission) filter sizes. The data from $n = 7$ fields of view are shown below. The overall pattern and mean values of SNR within manually-identified synaptic ROIs are very consistent from size filters ranging from 0 to $50 \times 50 \times 50 \text{ nm}^3$. At $50 \times 50 \times 50 \text{ nm}^3$, the mean SNR increases, likely due to the exclusion of dimmer puncta. At $60 \times 60 \times 60 \text{ nm}^3$, many puncta are excluded, bringing the effective number of sampled synaptic ROIs down, and the mean SNR increases even further. Given that our filter size is now $20 \times 20 \times 20 \text{ nm}^3$, as of

our second submission, we do not believe our choice of size filter greatly affected our results.

2) We re-ran the quantification of synaptic proteins within manually-identified beta-amyloid nanoclusters (**Fig. 4b**) with various size filters, chosen to span the range from no size filtration to double our current size filter size. The results shown below demonstrate that while the mean values decrease with increasing filter size, as expected, the overall pattern of protein abundance is consistent up to a filter size of ~60 voxels. We chose to proceed with a filter of 20 voxels, because this was the largest filter for which no amyloid-beta puncta, the basis of which these ROIs were manually identified, were excluded based on size.

3) We re-ran the calculation of the fraction of D54D2 volume mutually overlapped with GluA2 (**Fig. 4c**), with various size filters applied to both channels, ranging from 0 (no filter, which is what we report in the manuscript) to 180 voxels (over double what was used for the analysis in **Fig. 4b**). Results are shown in the plot below, with filter size on the x-axis. Data points from the same ROI over all filter

sizes are connected by black lines. The fraction of D54D2 volume mutually overlapped with GluA2 in the majority of ROIs does not change much by imposing a size filter especially for filter sizes <100 voxels (mean coefficient of variation of all filter sizes tested across ROIs = 0.1352, n = 71 ROIs from 9 fields of view from 2 5xFAD animals; note that only 44 of these ROIs, with low registration error, are shown in **Fig. 4c** as described in the methods). Thus, this result is quite robust to the choice of size filter parameter. We chose a size filter of 20 voxels to be consistent with the analysis in **Fig. 4b**.

Taken together, the results from the above tests suggest that within a certain range, the choice of size filter does not greatly affect our results or conclusions. Thus, aside from imposing a size filter of 20 voxels for the AMPA receptor-D54D2 colocalization analysis (**Fig. 4c-f**) for the sake of consistency with **Fig. 4b**, we have chosen not to change our use of these size filters in the manuscript. The figures, main text, methods, and Extended Data tables have been updated accordingly.

References

1. Shi, L. *et al.* Super-Resolution Vibrational Imaging Using Expansion Stimulated Raman Scattering Microscopy. *Adv. Sci.* **9**, (2022).
2. Alon, S. *et al.* Expansion sequencing: Spatially precise in situ transcriptomics in intact biological systems. *Science (80-.).* **371**, (2021).
3. Klevanski, M. *et al.* Automated highly multiplexed super-resolution imaging of

- protein nano-architecture in cells and tissues. *Nat. Commun.* **11**, (2020).
4. Jungmann, R. *et al.* Multiplexed 3D cellular super-resolution imaging with DNA-PAINT and Exchange-PAINT. *Nat. Methods* **2014 113 11**, 313–318 (2014).
 5. Lin, J. R., Fallahi-Sichani, M. & Sorger, P. K. Highly multiplexed imaging of single cells using a high-throughput cyclic immunofluorescence method. *Nat. Commun.* **2015 61 6**, 1–7 (2015).
 6. Lin, J. R., Fallahi-Sichani, M., Chen, J. Y. & Sorger, P. K. Cyclic Immunofluorescence (CyclF), A Highly Multiplexed Method for Single-cell Imaging. *Curr. Protoc. Chem. Biol.* **8**, 251–264 (2016).
 7. Lin, J. R. *et al.* Highly multiplexed immunofluorescence imaging of human tissues and tumors using t-CyCIF and conventional optical microscopes. *Elife* **7**, (2018).
 8. Ku, T. *et al.* Multiplexed and scalable super-resolution imaging of three-dimensional protein localization in size-adjustable tissues. *Nat. Biotechnol.* **2016 349 34**, 973–981 (2016).
 9. Valdes, P. A. *et al.* Improved immunostaining of nanostructures and cells in human brain specimens through expansion-mediated protein decrowding. *Sci. Transl. Med.* **16**, eabo0049 (2024).

Reviewers' Comments:

Reviewer #3:

Remarks to the Author:

The previous manuscript has many mistakes and typos, and Reviewer 1 and their student have pointed out. These improvements are necessary. One important point both Reviewer 1 and I are not very happy with is the statements related to the registration errors in this paper.

I appreciate that the authors now admit that “this method is limited by time-versus-resolution trade-off” and have added a note in the main text about this. What I don't appreciate is that the authors still claim the resolution of multiExR is 20nm (line 452, “MultiExR enables ~20 nm resolution imaging, with potentially very high multiplexing capacity”). They tried to soften the claim by adding a note at the beginning of the Discussion that MultiExR has a spatial resolution of 20nm or larger, which is not helping. This is almost like any conventional light microscope could claim its resolution is 200nm or larger. Expansion microscopy is a revolutionary invention, probably one of the most important technologies for light microscopy in the past decade, and has made a profound impact in the field. ExR is also a great innovation that can push the resolution up to 20nm. But, as I have mentioned in my previous comments, the resolution of MultiExR is not 20nm. Please don't make the conceptual shift. MultiExR is an extended pipeline for ExR to make it achieve multiplexing capacity.

Taking together with the 2nd comment from Reviewer 1, my suggestion is that the authors should honestly and rigorously state and discuss the quality of their method. The best way of further extending the advantages of expansion microscopy in biological research and helping researchers to have reasonable expectations and practical decisions in their studies, is to provide reliable pipelines with thorough and honest descriptions for MultiExR. The novelty of a stripping protocol and a registration software package can also be reflected in the fact that previously there were no existing reliable methods to achieve multiplexing for ExR. But, this method has to be reliable and rigorous enough to take the novelty claim. They claim the registration error is 10-40nm (Abstract), but even though 10nm is the lowest registration error, 40nm is the average registration error rather than the highest registration error in their datasets of Fig 2, 3, 4, and 5 (Extended tables). This claim is biased. I strongly suggest the authors either stop making any claims on the resolution of MultiExR but only state and discuss the “systematic registration error” of MultiExR in a conservative and honest way, or clearly and directly state the resolution of MultiExR is 60nm on average (20nm + 40nm).

Secondly, it's great that during the revision the authors found there was a bug in the data analysis of Fig 4. Indeed, the decision of the voxel threshold can be subjective depending on the questions the researchers are asking. I suggest the authors

incorporate this part of the rebuttal letters into the Extended Figures and the Methods part.

Lastly, the authors mentioned that they added information on NHS-ester staining as a reference channel into the Discussion but I can not find it. They did add a full paragraph to discuss the choice of reference channel, but not about the relationship between protocol duration and choice of reference channel, nor NHS-ester staining specifically. I also would suggest the authors add the line number to the rebuttal letter next to the changes they made.

We thank the reviewer for the comments and have done further writing to address all remaining issues in full. We hope the paper is now acceptable at Nature Communications.

Reviewer #3

The previous manuscript has many mistakes and typos, and Reviewer 1 and their student have pointed out. These improvements are necessary.

It is a bit unclear if the reviewer is referring to “mistakes” and “typos” in the version submitted in Feb 2024, which is the version that Reviewer 1 and their student were commenting on (please note - what Reviewer 1 stated was that the Feb 2024 version had fixed mistakes and typos since the previous version; curiously, none of the reviews of that previous version mentioned any mistakes or typos), or our last submitted version of May 2024, which was heavily revised since Reviewer 1 made those comments in Feb 2024. In any case, we have done multiple careful reads of the current manuscript, and do not think that there are obvious mistakes or typos. We of course welcome any specific critiques, and of course we defer to the editor regarding any remaining issues.

One important point both Reviewer 1 and I are not very happy with is the statements related to the registration errors in this paper.

I appreciate that the authors now admit that “this method is limited by time-versus-resolution trade-off” and have added a note in the main text about this. What I don’t appreciate is that the authors still claim the resolution of multiExR is 20nm (line 452, “MultiExR enables ~20 nm resolution imaging, with potentially very high multiplexing capacity”). They tried to soften the claim by adding a note at the beginning of the Discussion that MultiExR has a spatial resolution of 20nm or larger, which is not helping. This is almost like any conventional light microscope could claim its resolution is 200nm or larger. Expansion microscopy is a revolutionary invention, probably one of the most important technologies for light microscopy in the past decade, and has made a profound impact in the field. ExR is also a great innovation that can push the resolution up to 20nm. But, as I have mentioned in my previous comments, the resolution of MultiExR is not 20nm. Please don’t make the conceptual shift. MultiExR is an extended pipeline for ExR to make it achieve multiplexing capacity.

Taking together with the 2nd comment from Reviewer 1, my suggestion is that the authors should honestly and rigorously state and discuss the quality of their method. The best way of further extending the advantages of expansion

microscopy in biological research and helping researchers to have reasonable expectations and practical decisions in their studies, is to provide reliable pipelines with thorough and honest descriptions for MultiExR. The novelty of a stripping protocol and a registration software package can also be reflected in the fact that previously there were no existing reliable methods to achieve multiplexing for ExR. But, this method has to be reliable and rigorous enough to take the novelty claim. They claim the registration error is 10-40nm (Abstract), but even though 10nm is the lowest registration error, 40nm is the average registration error rather than the highest registration error in their datasets of Fig 2, 3, 4, and 5 (Extended tables). This claim is biased. I strongly suggest the authors either stop making any claims on the resolution of MultiExR but only state and discuss the “systematic registration error” of MultiExR in a conservative and honest way, or clearly and directly state the resolution of MultiExR is 60nm on average (20nm + 40nm).

We now have stopped using the word “resolution” in reference to multiExR, since it is ambiguous when used in the multiExR context. We utilize “resolution” to refer to ExR resolution as in our previous peer-reviewed paper on ExR, we utilize “registration error” to refer to the multi-round error of registration of multiExR, and we utilize “measurement error” to mean the combination of ExR resolution error and multiExR registration error. In more detail:

1. In the abstract, we modified our claim to “Across all datasets examined, multiExR exhibited a median round-to-round registration error of 39nm, with a median registration error of 25nm when the most stringent form of the protocol was used.” (lines 40-42). In the main text, we have elaborated: “In the most stringent form of the protocol (**Extended Data Fig. 1Bii**, path 2 in **Extended Data Fig. 1A**), which we applied to our primary validation dataset, we achieved a median (taken across all staining round pairs for all fields of view, i.e. each combination of round pair and field of view was considered as one sample) round-to-round registration error of 25 nm (minimum 14 to maximum 98 nm).” (Lines 90-94).
2. To give our readers a better idea of how multiExR registration error affects the ability to resolve the distances between biological objects (e.g. synaptic puncta), we have added a new section to the Discussion (lines 508-525) and a new **Extended Data Figure 7**:
“The appropriateness of multiExR for an experiment depends on the required measurement scale for the underlying biological question and the distribution of registration errors achieved for a given image. That is, what is the minimum distance between puncta (for example) that one needs to measure, below which two things are considered indistinguishable, and above which things are

considered separate? One useful observation is: registration error is not constant throughout an image, and thus regions can be found that enable higher precision measurements than others. For example, if 95% of the calculated registration error values fall within the range of 34-37nm (as for one round pair in Extended Data Table 6), then a user can with 95% confidence resolve distances 54-57nm in size (taking measurement error to be registration error + resolution). We provide information on the distribution of registration errors we achieved for each field of view in each round pair in **Fig. 2f**, **Fig. 3c**, **Extended Data Fig. 3b**, **Extended Data Fig. 4c**, and **Extended Data Tables 2, 4, 6, 11, and 14**. The software we provide calculates 1,000 estimates of registration error for every field of view (one for each randomly sampled subvolume above a signal threshold, see Methods), enabling the user to estimate the distribution of registration errors in the image. Should a reader want to examine only portions of the image that fall within a given registration error range (for example, 30nm or less), they can crop the image to regions with registration error in this range (**Extended Data Fig. 7**). Given the right-skew of the distributions of registration error that we found (**Fig. 2f**, **Fig. 3c**, **Extended Data Fig. 3b**, **Extended Data Fig. 4c**), the majority of the field of view could be in an acceptable range depending on the biological question at hand.”

To illustrate this procedure, we masked out an area of larger registration error in one field of view in the primary validation dataset (ROI1, rounds 1-4, registration error estimated distribution shown in **Fig. 2f**; **Extended Data Fig. 7a(i)-(ii)**). We qualitatively examined both a composite overlay of the synGAP channel (used to calculate registration error, **Extended Data Fig. 7a(i)**) and an RGB image showing the magnitude of the registration error in x, y, and z in red, green, and blue (**Extended Data Fig. 7b(ii)**), respectively to identify the area of higher registration error. Of note, the masked field of view retains ~80% of the original area in the x-y plane. Using this procedure, we were able to reduce the 95% confidence interval of the mean (across randomly sampled subvolumes within the field of view) registration error from 23.63-25.55nm to 18.24-19.57nm (**Extended Data Fig. 7c**) after outlier removal as described in the Methods). In theory, the registration error range could be reduced even further by more aggressive cropping to even lower registration-error regions.

Secondly, it’s great that during the revision the authors found there was a bug in the data analysis of Fig 4. Indeed, the decision of the voxel threshold can be subjective depending on the questions the researchers are asking. I suggest the authors incorporate this part of the rebuttal letters into the Extended Figures and the Methods part.

We have now added the description of the sensitivity of results to median and size filters in the Methods (lines 1063-1102) and a new **Extended Data Figure 8**.

We would also like to point out that the analysis which produced the results in **Fig. 4c** (overlap of GluA1-4 with beta-amyloid species) uses a 3D median filter of 5x5x3 voxels after binarization, which we did not describe in the text, even though it was in the code we supplied for review (via GitHub). For maximum clarity, we now mention this piece of code in the Methods (line 1106). For completeness, we re-ran the quantification of GluA1-4 and D54D2 volume overlap without a 3D median filter (i.e., size 0x0x0 voxels) to demonstrate the effect of the median filter on our results (**Extended Data Fig. 8d**).

Previously, we wrote that “...we found a significantly larger fraction of D54D2 containing GluA2 than GluA4 (**Fig. 4c**, $n = 44$ nanocluster ROIs from 8 fields of view from 2 5xFAD mice, **Extended Data Table 9(i-ii)** for statistics). We found essentially no GluA1 and very little GluA3 within D54D2 puncta (**Fig. 4c**.” Eliminating the median filter did not affect the pattern of which AMPAR subunits had the most colocalization with D54D2, but did increase the numerical value of the normalized mutually overlapped volume, especially in the case of GluA3.

We also wrote: “Additionally, there were many more ROIs for which there was no GluA4 contained within a D54D2 punctum, than for GluA2 (**Fig. 4c**, 4.55% zero for GluA2 vs. 52.3% for GluA4; Chi-squared test, Chi-square = 24.64, $p < 0.0001$, $n = 44$ nanocluster ROIs from 8 fields of view from 2 5xFAD mice).” With no median filter, there were still more ROIs for which there was no GluA4 contained within a D54D2 punctum than for GluA2 (0% for GluA2 vs. 22.7% for GluA4; Chi-square = 11.28, $p = 0.0008$, $n = 44$ nanocluster ROIs from 8 fields of view from 2 5xFAD mice).

Regarding “Leveraging the multiplexed nature of the data, we performed pairwise linear regressions on the volume of GluA4 and GluA2 vs. D54D2 present in A β nanocluster ROIs, and found that both were highly correlated, but the best-fit line for GluA2 vs. D54D2 volume was shifted up from that of GluA4 vs. D54D2 (**Fig. 4d**, **Extended Data Table 9(ii)** for full statistics).” - with no median filter, this statement is still true.

Regarding “For A β nanocluster ROIs in which GluA4 was present, the volume of GluA2 present was correlated with that of GluA4 (**Fig. 4e**, **Extended Data Table 9(iii)** for full statistics). Visual inspection of GluA2 and GluA4 in A β ROIs chosen from different parts of the distributions of **Fig. 4c** illustrate these observations (**Fig. 4f**.” - with no median filter, this statement is still true. Indeed, the correlation was slightly higher ($R^2 = 0.7099$, $p < 0.0001$) due to fewer zero values for GluA4.

In short, none of the conclusions were affected by the use of the median filter.

Lastly, the authors mentioned that they added information on NHS-ester staining as a reference channel into the Discussion but I can not find it. They did add a full paragraph to discuss the choice of reference channel, but not about the relationship between protocol duration and choice of reference channel, nor NHS-ester staining specifically. I also would suggest the authors add the line number to the rebuttal letter next to the changes they made.

Regarding “the authors mentioned that they added information on NHS-ester staining as a reference channel into the Discussion but I can not find it”: we apologize for referring to a change in the text that did not make its way into the final last submitted version. We have added a paragraph to the discussion section (lines 526-534).

Regarding “They did add a full paragraph to discuss the choice of reference channel, but not about the relationship between protocol duration and choice of reference channel”: we discuss the speed vs. resolution tradeoff in lines 491-499 and lay it out in visual form in **Extended Data Fig. 1**. We have added the following sentence to relate this to the choice of reference channel: “There is a relationship between choice of reference channel and protocol duration: dedicating more optical channels to reference stains improves registration quality at the cost of the number of target stains that can be imaged per round”.

Regarding “I also would suggest the authors add the line number to the rebuttal letter next to the changes they made”, we have now done so.

Reviewers' Comments:

Reviewer #3:

Remarks to the Author:

The authors have addressed my concerns. I have no further comments.